# Right Ventricle and Epigenetics: A Systematic Review

**DOI:** 10.3390/cells12232693

**Published:** 2023-11-23

**Authors:** Victoria Toro, Naomie Jutras-Beaudoin, Olivier Boucherat, Sebastien Bonnet, Steeve Provencher, François Potus

**Affiliations:** Centre de Recherche de l’Institut Universitaire de Cardiologie et de Pneumologie de Québec (CRIUCPQ), Québec, QC G1V 4G5, Canada; victoria.toro.1@ulaval.ca (V.T.); naomie.jutras.beaudoin@usherbrooke.ca (N.J.-B.); olivier.boucherat@criucpq.ulaval.ca (O.B.); sebastien.bonnet@criucpq.ulaval.ca (S.B.); steeve.provencher@criucpq.ulaval.ca (S.P.)

**Keywords:** epigenetic, right ventricle, DNA methylation, microRNA, histone, lncRNA, pulmonary hypertension, tetralogy of Fallot, fibrosis, cardiomyopathy, preclinical model, systemic ventricle, arrhythmogenic

## Abstract

There is an increasing recognition of the crucial role of the right ventricle (RV) in determining the functional status and prognosis in multiple conditions. In the past decade, the epigenetic regulation (DNA methylation, histone modification, and non-coding RNAs) of gene expression has been raised as a critical determinant of RV development, RV physiological function, and RV pathological dysfunction. We thus aimed to perform an up-to-date review of the literature, gathering knowledge on the epigenetic modifications associated with RV function/dysfunction. Therefore, we conducted a systematic review of studies assessing the contribution of epigenetic modifications to RV development and/or the progression of RV dysfunction regardless of the causal pathology. English literature published on PubMed, between the inception of the study and 1 January 2023, was evaluated. Two authors independently evaluated whether studies met eligibility criteria before study results were extracted. Amongst the 817 studies screened, 109 studies were included in this review, including 69 that used human samples (e.g., RV myocardium, blood). While 37 proposed an epigenetic-based therapeutic intervention to improve RV function, none involved a clinical trial and 70 are descriptive. Surprisingly, we observed a substantial discrepancy between studies investigating the expression (up or down) and/or the contribution of the same epigenetic modifications on RV function or development. This exhaustive review of the literature summarizes the relevant epigenetic studies focusing on RV in human or preclinical setting.

## 1. Method

### 1.1. Search Strategy and Study Selection

All the published data were searched and collected between the inception of the study and 1 January 2023, according to the preferred reporting items for systematic reviews and meta-analyses (PRISMA) recommendations for systematic reviews [1]. The electronic search was conducted on PubMed and designed to capture maximum sensitivity for the right ventricle (RV) and epigenetic modification (DNA methylation, histone modification, long non-coding RNA, microRNA, and chromatin modification). A comprehensive description of the search strategy is available in the Appendix A. The following inclusion criteria were used to obtain more specific search results: original research articles accepted or published with availability in electronic databases by 1 January 2023, and articles only in English. Review articles and conference abstracts were excluded. The articles that did not directly focus on epigenetic modifications and the RV in healthy, developmental, and pathological conditions were excluded.

### 1.2. Data Extraction

After the search, two independent examiners (V.T. and F.P.) screened and reviewed the research titles, abstracts, and main text. When discordant, records were discussed by the evaluators and included/excluded when reaching a consensus. The abstracted information included the epigenetic modification, the pathology, human patients’ characteristics (sample size, age, sex, tissues/sample type), in vivo (sample size, age, sex, species, model) and in vitro (cell type) models, and treatments or therapeutical intervention and outcomes. The data collection, management, and analysis of all relevant evidence for epigenetic modifications and RV are presented in the flow diagram (Appendix A).

### 1.3. Objectives

The first objective of this systematic review is to provide an unbiased and exhaustive overview of the current knowledge on the epigenetic modification (DNA methylation, histone modification and non-coding RNA) associated with physiologic/pathological RV development and function/dysfunction. The second is to summarize the epigenetic modifications translated into clinical/preclinical therapeutic interventions and biomarkers for RV function/dysfunction. The third is to highlight the potential concordance/discrepancy between the studies focusing on the same epigenetic modification.

## 2. Introduction

The right ventricle (RV) is a thin-walled crescent structure of the heart, connecting the systemic venous return to the pulmonary circulation. In the physiologic condition, the RV works in a low-pressure system that receives venous blood from the atrium through the tricuspid valves and expulses it to the pulmonary artery through the outflow tract. The RV shape and pressure stress make it significantly different to the cone-shaped left ventricle (LV), which is adapted for a high-pressure system. Moreover, RV and LV have distinct embryologic origins. The RV (and outflow track) emerges from the second heart field whilst the LV, the muscular interventricular septum, and the atria rise from the first heart field [2]. Consequently, the RV has a more distinct transcriptomic signature than the LV or atria [3,4]. This specificity precludes any extrapolation of the knowledge from the LV to the RV.

In terms of physiopathology, RV dysfunction is associated with ischemic, non-ischemic (e.g., arrhythmogenic cardiomyopathy), congenital (e.g., tetralogy of Fallot (TOF), pulmonary stenosis), and arrhythmogenic cardiomyopathy (ACM). Moreover, RV dysfunction and failure are often the culminating point of chronic lung (e.g., chronic obstructive pulmonary disease (COPD) and pulmonary hypertension (PH), amongst others) and left heart diseases. Whatever the underlying condition, impaired RV function is almost systematically associated with patients’ decreased quality of life and increased morbidity and mortality [5,6,7]. However, until recently, the RV received far less attention than the LV and most molecular knowledge on RV dysfunction was extrapolated from the LV.

Functionally, RV dysfunction and failure are associated with drastic changes involving, among others, cardiomyocyte hypertrophy, metabolic reprogramming, increased fibrosis and inflammation, and impaired angiogenesis [8]. How and what orchestrates those pathological modifications remains nebulous; but epigenetic regulation of gene expression has emerged as a critical determinant of embryogenesis, cardiovascular diseases, LV hypertrophy, and left heart failure [9]. Epigenetics is defined as a “stable heritable phenotype resulting from changes in a chromosome without alterations in the DNA sequence”. It results in the modulation of gene expression in response to the environment, such as pathologic or physiologic stressors, through: (1) DNA methylation, (2) post-translational modifications of histones, and (3) non-coding RNAs [10,11].

Although less investigated, recent studies pinpointed the potential contribution of epigenetics to the physiologic and pathologic regulation of RV function. The role of epigenetics in RV development and the progression of RV dysfunction, its capacity to predict RV dysfunction, and the potential of epigenetic-based therapy remain largely elusive. We thus performed a systematic exhaustive review of the literature and to summarize the relevant epigenetic studies focusing on the RV in humans and animal models.

## 3. Results

Our literature search retrieved 817 studies that were assessed for eligibility. We excluded 708 studies and included 109 separate publications reporting epigenetic modifications. The reasons for excluding studies appear in Appendix A. The included studies explored epigenetic modifications related to non-coding microRNAs (*n* = 65, Table 1), long non-coding RNA (lncRNA, *n* = 12, Table 2), circular RNA and small nucleolar RNA (*n* = 3, Table 3), histones modification (*n* = 15, Table 4), and DNA methylation (*n* = 17, Table 5).

### 3.1. Micro-RNA

Non-coding RNA with a length of <22 nucleotides are classified as micro-RNA (miRNA). miRNAs are, by far, the most studied epigenetic modifications in RV dysfunction (Table 1). The canonical miRNA biogenesis includes three sequential steps [12]. The miRNAs are transcribed from DNA sequences into primary miRNAs, processed into precursor miRNAs, and then into mature miRNAs. Functionally, mature miRNAs act as posttranscriptional regulators of gene expression. In most cases, miRNAs interact with target messenger RNAs (mRNAs) to induce mRNA degradation and/or translational repression. Notably, miRNAs can activate translation or regulate transcription in a non-canonical pathway [13]. miRNAs can also be secreted into extracellular fluids and transported to target cells via vesicles, such as exosomes, or by binding to proteins. Interestingly, a phylogenic study conducted in healthy rats, dogs, and monkeys showed that the miRNA profile is relatively well conserved in the RV, compared to the apex, septum, LV, and papillary muscle, but is poorly conserved across the species [14]. Not surprisingly, miRNAs have been repeatedly associated with diverse conditions leading to RV dysfunction.

**Table 1 cells-12-02693-t001:** microRNAs and right ventricle.

Epigenetic Modification	Human	In Vivo	In Vitro	Therapeutic Intervention	Outcomes	PMID/Ref
**Arrhythmogenic Cardiomyopathy**						
**miR-320a**	Plasma: - 53 ctrl (53% male, 42.9 y +/− 1.57) - 21 IVT (21% male, 48.1 y ± 3.09), - 36 ACM (100% male, 48.1 y ± 2.22)	N/A	N/A	N/A	- ACV vs. ctrl and IVT => ↓ miR-320a - miR-320a levels don’t correlate with ACM severity	28684747/[15]
**miR-130a**	N/A	Mice, CD-1, αMHC-tTA/TetO-miR130a	3T3	- Myocyte specific miR-130a overexpression - miR-130a inhibitor	In vivo: - ↑ αMHC miR-133a => ↑ RV hypertrophy and arrhythmogenic - ↑ miR-133a => fibrosis, Lipid, accumulation, CM death in LV, RV. In vitro: - miR-130a => direct regulator of *DSC2*	27834139/[16]
**- Cardiac related Array (84 miR)**	RV: - 9 ACM - 4 ctrl Blood: - 9 ACM (discovery) - 90 ACM (validation) - 24 ctrl	N/A	N/A	N/A	RV and blood (discovery cohort): - ACM vs. ctrl, differentially expressed miR in a consistent direction => miR-122-5p, miR-133a-3p, miR-133b, miR-142-3p, miR-144-3p, miR-149-3p, miR-182-5p, miR-183-5p, miR-208a-3p, miR-122-5p, miR-133a-3p, miR-133b, miR-142-3p, miR-144-3p, miR-149-3p, miR-182-5p, miR-183-5p, miR-208a-3p, miR-494-3p. - ACM vs. ctrl, blood (validation cohort): - ↑ miR-122-5p, miR-182-5p, and miR-183-5p. - ↓ miR-133a-3p, miR-133b, miR-142-3p.	32102357/[17]
**- microarray**	RV: - 24 ARVC (66% male, 36 y +/− 13) - 24 ctrl	N/A	N/A	N/A	- ↑ miR-21-3p, miR-21-5p *, miR-34a, miR-212, miR-216a, miR-584, miR-1251, miR-3621miR-3674, miR-3692, miR-4286, miR-4301 in ARCV vs. ctrl - ↓ miR-135b *, miR-138, miR-193b, miR-302b, miR-302c, miR-338, miR-451a, miR-491, miR-575, miR-3529, miR-4254, miR-4643 in ARCV vs. ctrl * correlate with Wnt and Hippo pathways, as well as myocardium adiposis and fibrosis.	27307080/[18]
**- qRT-PCR (754 miRNAs)**	Plasma: - 37 ARVC (59% male, 44 y ± 13), - 30 ctrl (age sex matched)	N/A	N/A	N/A	- ARVC vs. ctrl => ↑ miR-185-5P	34685557/[19]
**- RNAseq**	Pericardial fluid - 6 ARVC (22% female, 37 y +/− 15.8), - 3 post-infarction VT (100% male, 65 y +/− 10.8)	N/A	N/A	N/A	- ARVC vs. post infarction VT => miR-1-3p, miR-21-5p, miR-122-5p, miR-206, miR-3679-5p differentially expressed	33816578/[20]
**- hsa-let-7e, miR-122-5p, 133a, 144-3p, 145-5p, 185-5p, 195-5p, 206, 208a, 208b, and 494**	Plasma: - 28 ARVC (60% male, 48 y +/− 13) - 11 borderline ARVC (45% male, 48 y +/− 19) - 23VT (34% male, 40 y +/− 10) - 33 ctrl	N/A	N/A	N/A	- ↑ miR-144-3p, 145-5p, 185-5p, and 494 in ARVC with VA - ↑ miR-494 ARVC with recurrent VA post ablation.	29036525/[21]
**- RNA seq** **- q RT-PCR**	Blood - 9 chronically paced (46% male, 15.7 y +/ 2.4) - 13 non-paced ctrl (33% male, 15.03 z +/− 2)	N/A	N/A	N/A	- Paced vs. ctrl => ↓ 192, ↑ 296 miR - PCR validation selected miR paced vs. ctrl => ↑ miR-214-3p, miR-210-5p, and miR-205-5p, ↓ miR-130b-5p, miR-190a-5p, miR-148b-5p, miR-126-5p, and miR-15b-3p.	36121621/[22]
**Coronary artery disease—cardiac allograph vasculopathy**						
**- miR-126** **- miR-628-3p** **- miR-92a-3p**	RV - 21 cardiac allograph vasculopathy (CAV+, 85% male, 51.1 y +/− 11.8) - 18 CAV− (89% male, 59.9 y +/− 7) - 8 end stage CAD	N/A	N/A	N/A	- CAV+ vs. CAV− => ↓ miR-126-3p - ↓ miR-126-3 predicts CAV event - CAD vs. CAV− => ↑ miR-126-3p - miR-628-30 and miR-92a-3p levels unaffected across the conditions	32548243/[23]
**Congenital heart disease**						
**- RNAseq**	RVOT: - 13 CHD (54% males, 1.8 y ± 1.69) - 7 controls (43% males, 25.29 W ± 1.70)	- Zebrafish embryo	- HL1	- Mimic-miR-29b - Inhibitor-miR-29b	Human: - CHD vs. ctrl => impaired expression of 20 miRNA - CHD vs. ctrl => ↑ miR-29b-3p In Vivo/Vitro - ↑ miR-29b-3p induce cardiac malformation and lethality in zebra fish, - ↑ miR-29b-3p inhibited cardiomyocyte proliferation in vitro and in vivo - ↓ miR-29b-3p ↑ cardiomyocyte proliferation in vitro and in vivo. - miR-29b targets *NOTCH2*	32077168/[24]
**- miR-486** **- micro array (cells)**	RV - 3 HLHS (newborn) - 3 controls (newborn)	- sheep - aortopulmonary vascular graft	- EMCM - Cyclic stretch	In vitro: - mimic-miR-486	Human - HLHS vs. ctrl => ↓RV miR-486 In Vivo - hypertrophic RV => ↓RV miR-486 In vitro - Cyclic stretch => impairs expression of 34 miRNA. - Cyclic stretch => ↓RV miR-486 - ↑ miR-486 => ↑ contractility - ↑ miR-486 => ↓ *FoxO1*, ↓ *Smad*, ↑ *Stat1*.	31513548/[25]
**Dilated cardiomyopathy**						
**- Microarray**	RV: - 8 End stage DCM with LVAD (87.5% male, 57 y [26,27,28,29,30]), - 8 DCM without LVAD, (87.5% male, 50 y [26,27,28,29,30,31,32,33,34,35,36,37]) - 6 no HF (45 y, [26,27,28,29,30,31,32,33,34,38,39,40], no info sex)	N/A	N/A	N/A	- DCM no LVAD vs. no HF => 19 differentially expressed miRNAs. - 3 miRNAs (hsa_miR_21 *, hsa_miR_1972 and hsa_miR_4461) in the RV showed normalization after LVAD implantation	35216165/[41]
**- miR-21,** **- miR-26,** **- miR-29,** **- miR-30** **- miR-133a.**	Plasma - 15 DCM with normal RV (80% male, 45.6 y +/− 12.1) - 55 DCM + RVD/SD (7.3% male, 48.7 y +/− 12.1)	N/A	N/A	N/A	- DCM + RVD/SD vs. DCM => ↑ miR-133a - MiR-21, miR-26, miR-30, and miR-133a correlate with RV morphological but not with functional parameters. - MiR-30 associated with RV impairment	28840590/[42]
**- miR-29a** **- miR-29b** **- miR-29c** **- miR-133a** **- miR-133b**	Apex, LV, septum, RV tissues. - 3 DCM (46.57 y ± 9.06) - 3 ctrl (32.0 y ± 7.16)	N/A	N/A	N/A	- DCM vs. ctrl RV => ↓ miR-29b, -133a, -133a, -29c. - DCM vs. ctrl RV => ↑ miR-21	27922664/[43]
**- miR-21** **- miR-26** **- miR-29** **- miR-30** **- miR-133a**	Plasma and RV - 70 DCM, 48.04 y ± 12.1 - 7 patients with CAD who underwent CABG (ctrl), 72.5 y ± 3.4	N/A	N/A	N/A	- DCM vs. ctrl RV => ↓ miR-133a, ↓ miR-26, ↑ miR-29 - DCM vs. ctrl plasma => ↓ miR-26, ↑ miR-29 - miR-133a RV => independent predictor of mortality	29377565/[44]
**- miR-21** **- miR-26** **- miR-29** **- miR-30** **- miR-133a** **- miR-423**	Plasma and RV - 32 DCM + LVRR, 48.9 y ± 9.9, 9.4% female - 31 DCM without LVRR, 46.9 y ± 9.9, 9.7% female	N/A	N/A	N/A	- DCM LVRR vs. DCM no LVRR => ↑ miR-133a RV - miR-133a => independent predictor of DCM LVRR	32207584/[45]
**Irradiation**						
**- miR-21** **- miR-1**	N/A	- Rats, Wistar, Male - Single dose of 25 Gy of ionizing radiation	N/A	In vivo - Aspirin, - Atorvastatin - sildenafil	- no changes in RV miR-1 post irradiation. - ↑ RV mi-R21 post irradiation. - aspirin, atorvastatin, and sildenafil => ↓ RV miR-21 post irradiation	29642568/[46]
**Neonatal Hypoxia**						
**- miR-206** **- miR-1**	N/A	- Rats, Sprague Dawley, male+ female (2 days) - Hx 12% (2–20 days)	N/A	N/A	- hypertrophic RV => ↓ miR-206 - hypertrophic RV => no differences in miR-1 expression	23842077/[47]
**Normal heart**						
**-microarray**	N/A	- Rats, Wistar Han, male, (3–6 months) - Dogs, Beagle, male, (25–37 months) - Monkeys, cynomolgus, female, (2–4 years).	N/A	N/A	- Apex, septum, RV, LV, and papillary muscle => similar miRNA profile. - Heart miRNA profile poorly conserved across the species.	23300973/[14]
**Pulmonary Hypertension**						
**- microarray**	N/A	- Dog, male, mongrel (9–13 m, 21–27 kg) - Tachypacing-induced biventricular HF - group 2 PH model.	- DRVF - Cyclic overstretch - Aldosterone	IN vitro: - AntagomiR-21	- HF vs. ctrl => ↑ miR21, miR221 in RV (not in LV). - In vitro miR21 and 221 increased in stressed DRVF	31916447/[48]
**- PCR, 93 mir**	N/A	- mixed-breed Western ewes, 137−141 days gestation - Fetal aortopulmonary shunt - haemodynamic measurement => 4–6 w	N/A	N/A	- Fetal shunt vs. sham RV => ↑ 40 miRNA. - ↑ miR-199b might contribute to the regulation of Dyrk1a/NFAT pathway - ↑ miR-29a contributes to fibrosis	29906222/[49]
**- PCR array (753 miR)**	Plasma, from superior vena cava, pulmonary artery, and ascending aorta: - 12 PAH (42% male, 7.8 y (0.5–17.7)), - 9 ctrl (56% male, 6.5 (0.4–17.1))	N/A	N/A	N/A	- Trans RV miRNA gradients (pulmonary artery vs. superior vena cava): miR-193a-5p (↑ in PAH, ↓ in ctrl) and miR-423-5p (↓ in PAH and ctrl) - transpulmonary miRNA gradients (ascending aorta vs. pulmonary artery): miR-26b-5p (↓ in ctrl), miR-331-3p (↑ in PAH). - miR-193a-5p, miR-423-5p, miR-26b-5p, miR-331-3p correlate with haemodynamic - PAH vs. ctrl ↑ miR-29a-3p, miR-26a-5p, miR-590-5p, and miR-200c-3p in PAH-superior vena cava and ↓ miR-99a-5p in PAH -pulmonary artery	31876555/[50]
**- myomiRs (miR-1, miR-133a, miR-208, miR-499) and miR-214**	N/A	- Rats, Wistar, Male - MCT, 60 mg/kg, s.c.	N/A	N/A	- MCT vs. Ctrl =>↓ miR-208a, ↓miR-1, ↓miR-133a and ↓miR-499 in RV - MCT vs. Ctrl => ↑ cardiac damage-related miR-214 in failing RV.	32395887/[33]
**- Mir-21**	plasma, group 3 PH, *n* = 41 - 19 COPD, - 9 bronchiectasis, - 7 pulmonary tuberculosis, - 6 IPF. RV dysfunction: - 31 without RV dysfunction (67.7% male, 56.2 y ± 8.5), −10 with RV dysfunction (60% male, 51.6 y ± 19.4)	N/A	- H9C2	- Mimic-miR21	Human: - Circulating miR-21correlated with RV dysfunction In Vitro: - miR-21 => ↑ CM hypertrophy - miR-21 => ↑ cardia stress markers (*BNP*) expression.	30449992/[51]
**- Mir-1**	N/A	- Rats, Sprague-Dawley, Male (180–200 g) - Hx (10% O_2_, 4 w)	- RCF - Hx	in vitro/in vivo: - mimic-miR1 - antagomiR-miR-1	In Vivo: - Hx RV => ↑ miR1 - ↓ miR1 => ↓ RV hypertrophy, - ↓ miR1 => ↓RV fibrosis In Vitro: - Hx RCF => ↑ miR1 - ↓ miR1 => ↓ collagen production	33604679/[37]
**- Mir-495**	N/A	- Rats, Sprague-Dawley, Male - MCT, 60 mg/kg, s.c.	- NRVC	In Vitro: - mimic-miR495 - antagomir-miR495	In Vivo: - ↓RV miR-495 => ↑ *PTEN*. In vitro: - ↑ miR-495 prevents CM hypertrophy - ↑ miR-495 => ↓ cardiac stress marker (e.g. *NPPA*)	29566365/[34]
**- Mir-322 (424)**	N/A	- Rats (6–8 w) - MCT, 40 mg/kg, s.c.	- C1C12 (mouse myoblast)	In Vitro - Mimic-miR322 (424) - Antagomir- miR322 (424)	In Vivo - MCT vs. ctrl RV => ↓ Mir-322 (424) In Vitro - Mir-322 (424) directly targets *IGF-1*	29511611/[28]
**- Mir-322 (424)**	Plasma: - 14 IPAH, HPAH, DPAH (54 y, 40–67; 80% F) - 15 PAH-CTP (59 y, 51–71; 80% F) - 32 PAH CHD (37 y, 31–47; 59% F) - 3 PoPH (52 y, 31–67; 100% F) - 24 CTEPH (60 y, 49–73; 58% Female) - 34 healthy control (69 y, 64–75; 52% F)	- Rats, Wistar, Male - MCT, 60 mg/kg, i.p.	- NRVC - H9c2	N/A	Human: -PH patients vs. Ctrl = ↑ miR-424(322) - miR-424(322) corelates with symptoms and hemodynamics - miR-424(322) predicts survival in CHD-PH. In Vivo: - association between circulating miR-424(322) levels and RV hypertrophy - RV miR-424(322) correlates with ↓ *SMURF1* expression In Vitro: - hypoxia induces the secretion of miR-424(322) by PAECs, which after being taken up by cardiomyocytes leads to down-regulation of *SMURF1*.	29016730/[29]
**- Mir-325**	N/A	- Rats, Sprague-Dawley, Male - MCT, 60 mg/kg, i.p..	-RCF - exposed to angII	In Vivo - Lentivirus-miR-325-3p. In vitro - Mimic-miR-325-3p	In Vivo: - MCT vs. Ctrl RV => ↓ miR-325 - miR-325 negatively correlates withRV fibrosis - ↑ miR-325 => HE4 => ↓ RV fibrosis In Vitro - angII => ↓ miR-325 - ↑ miR-325 => ↓ fibrosis	35136419/[52]
**-Mir-223**	N/A	- Mice, C57BL/6, Male - Hx (10% o2, 21d), -PAB, - transgenic miR-223 knock-out mice.	- cos-7	In vivo: - miR-223 genetic depletion - adeno-associated virus cardiac specific pre-miR-223 overexpression - AntagomiR-223 In vitro - AntagomiR-223	In Vivo: - Hx => ↓miR-223 - PAB => ↓ miR-223 - ↑ miR-223 => ↓ RV fibrosis, ↑ RV function - ↓ miR-223 => ↓ RV function - miR-223 targets *IGF-IR*	27013635/[26]
**- Mir-214** **- Mir-199**	N/A	- Rat, Wistar Kyoto, - mice, C57BL/6, Male + female, 8 weeks - C57BL/6 depleted for miR-214 - SUHX, 20 mg/kg, s.c.	N/A	miR-214 genetic depletion	- SuHX vs. ctrl (rats and mice) => ↑ mir-214, ↑ miR-199 - ↓ miR-214 => ↑RV hypertrophy - miR-214 targets phosphatase and *tensin* and *PTEN*	27162619/[30]
**- Mir-21**	Blood, - 19 CHD-PAH + HF (50 y +/− 23.7; 37% male) - 57 CHD-PAH no HF (52 y +/− 22; 33% male) 10 healthy controls (50 y +/− 8; 40 % male)	- Rats, Sprague-Dawley, Male - aorto-venous fistula	- H9C2 cardiomyocytes, - microflow-mediated shear stress	In vitro -Mimic-miR-21	Human - CHD-PAH HF vs. no HF => ↓ miR-21 - Multivariate analysis => miR-21 levels associated with HF hospitalization. In Vivo - ↑ RV mir-21 RV in early response to aeorto-venous fistula - ↓ RV mir-21 RV in late response to aeorto-venous fistula In vitro - ↑ miR-21 in cardiomyocytes under shear stress at 3 h and ↓ at 6 h. - ↓ miR-21 => ↑ apoptosis. - ↑ miR-21 prevent CM apoptosis.	35159373/[38]
**- Mir-21**	N/A	- Rats, Sprague-Dawley, Male - SUHX, 20 mg/kg, s.c., 10% O_2_	- Adult RRVC - Adult RLVC - Hx	In vivo - Trimetazidine (rat 10 mg/kg/days, 4 weeks) In vitro - Trimetazidine (10 µM)	In vivo - ↓ miR-21 in decompensated RV - Trimetazidine => ↑miR-21 expression => ↑ RV function In Vitro - ↓ miR-21 => ↑ RRVC apoptosis - Trimetazidine => ↑miR-21 => ↓ apoptosis	22842854/[39]
**- Mir-208**	N/A	- Rats, Sprague-Dawley, Male - MCT, 60 mg/kg, s.c.	- Adult RRVC - Adult RLVC - Hx - NRC - TNF and phenylephrine	In vitro -mimic-miR-208 - antagomir-208	In vivo - ↓ miR-208 in decompensated RV. In vitro - miR-208 and Inflammation regulates the MED13/NCoR1-Mef2 Axis in the RV but not in the LV - ↓ mir-208 inhibition decreases cardiomyocyte hypertrophy	25287062/[53]
**- Mir-200b**	N/A	- Mice, C57BL/6, Male - MCT 60 mg/kg, i.p.	N/A	In Vivo - mimic-miR-200b	- ↓ miR-200b in MCT RV - ↑ miR-200b => ↓ RV *PKCa* expression	31730233/[54]
**- microarray (miR-197, miR-146b, miR-133, miR 491)**	RV - 7 IPAH - 6 Ctrl	- Rats, Sprague-Dawley, Male - SUHX, 20 mg/kg, s.c., 10% O_2_	- NRCM	In Vivo - pioglitazone In vitro - pre-miR-197 - pre miR-146b - pioglitazone	Human - IPAH vs. Ctrl RV => ↑ miR-197, ↑ miR-146b In vivo - SuHx vs. Ctrl RV => ↑ miR-197, ↑ miR-146b, ↓miR-133, ↓miR 491 - pioglitazone => ↓miR-197, ↓ miR-146b and ↑miR-491 In Vitro - ↑ miR-197, ↑ miR-146b => ↓ genes that drive FAO (*CPT1B, FABP4).*	29695452/[27]
**- miR-17** **- miR-21** **- miR-30b** **- miR-145** **- miR-204** **- miR-424** **- miR-503**	Plasma, - 14 PAH - 13 controls	- Rats, Sprague-Dawley, Male, SUHX, 20 mg/kg, s.c., 10% O_2_ - Rats, Fisher, Male, MCT, 60 mg/kg, i.p. - Mice, C57BI6J, male, Hypoxia (10% o2)	N/A	N/A	Human - PAH vs. Ctrl => ↓ miR-17, ↓ miR-145, ↑ miR-424 In vivo - Rat MCT vs. ctrl RV => ↓ miR-145. - Rat SuHX vs. ctrl RVI => ↑ miR-21 and ↓ miR-204 - Mice Hx vx NX RV =>↑ miR-322 and ↑ miR-503.	25763574/[55]
**- miR-17-5p** **- miR-21-5p** **- miR-126-3p** **- miR-145-5p** **- miR-150-5p** **- miR-204-5p** **- miR-223-3p** **- miR-328-3p** **- miR-424-5p**	N/A	- Rats, Sprague Dawley, Female - MCT, 60 mg/kg, i.p.	N/A	In vivo: - AntagomiR-223	- MCT vs. ctrl RV => ↑ miR-17, ↑ miR-21, ↑ miR-223, ↑ miR-503 - MCT vs. ctrl RV => ↓ miR-145, ↓ miR-150, ↓ 424 - treatment did not changed RV miR-223 expression	26815432/[40]
**-miR-126**	RV free wall tissue - 17 Ctrl (62 +/− 4 years, 53% female) - 8 CRV (40 +/− 7 years, 50% female); - 14 PAH DRV (53 +/− 4 years 79% female)	- Rats, Sprague-Dawley, Male - MCT, 60 mg/kg, s.c.	- Human EC from ctrl, CRV and DRV	In vivo and in vitro: - mimic-miR-126 - antagomiR-126	Human: - ↓ miR-126 in DRV In vivo - ↓ miR-126 in DRV - ↑ miR-126 in RV => ↑ capillary density => ↑ RV function In vitro - ↓ miR-126 in EC from DRV - ↑ miR-126 => ↑ EC angiogenic potential - ↑ miR-126 => ↓ *SPRED-1* => ↑ angiogenesis	26162916/[56]
**-miR-1**	N/A	- Rats, Sprague-Dawley, Male - MCT, 60 mg/kg, s.c.	- human LHCN-M2 skeletal myoblasts	In vitro - mimic-miR-1	In vivo - MCT vs. ctrl RV => ↓ miR-1 In vitro - miR-1 targets *TGF-βR1* and reduces *TGF-β* signalling	32109943/[36]
**- microarray**	N/A	Rats, Sprague-Dawley, Male - SUHX, 20 mg/kg, s.c., 10% O_2_	N/A	N/A	In vivo - SuHX vs. ctrl RV => ↑ miR-21-5p, ↑ miR-31-5 and 3p, ↑ miR-140-5 and 3p, ↑ miR-208b-3p, ↑ miR-221-3p, ↑ miR-222-3p, ↑ miR-702-3p, ↑ miR-1298 - SuHX vs. ctrl RV => ↓ miR-187-5p, ↓ miR-208a-3p, ↓ miR-877 - miR-140 expression correlates with *mitofusin-1* expression in PH RV.	27422986/[31]
**- microarray**	N/A	Rats, CRV - PAB - chronic hypoxia (10% O_2_, 4 weeks) Rats DRV - SUHX, 20 mg/kg, s.c., 10% O_2_ -PAB + low copper diet	N/A	N/A	In Vivo, - DRV vs. CRV => ↓ miR-133a, miR-139-3p, miR-21 and miR 34c - Normal RV and LV => discernable differences in miRNA expression	21719795/[57]
**-750 miRNAs by qPCR arrays**	RV - 6 Ctrl - 4 IPAH (25 y +/− 11.5, 75% female)	Mice, FVB, - Hx10% O_2_, 18 h, 48 h, 5 days	- NRC - Hx	In vitro: - premiR-146b - AntagomiR-146b	In vivo: - Hx regulated miR in RV and LV => let-7e-5p, miR-29c-3p, miR-127-3p, miR-130a-3p, miR-146b-5p, miR-197-3p, miR-214-3p, miR-223-3p, and miR-451 - ↓ miR-146b in Hx RV In vitro - Hx => ↓ miR-146b - ↓ miR-146b => ↑ *TRAF6,* and ↑ *IL-6* and ↑ *CCL2(MCP-1)*	31338525/[58]
**- miR-20a-5p** **- miR-17-5p** **- miR-93-5p** **- miR-3202** **- miR-665**	Blood - 8 CTEPH (50% female, 61 y +/− 6.8)	N/A	N/A	N/A	- miR-20a-5p correlates with all the RV function echocardiographic parameters - miR-93-5p, miR-17-5p and miR-3202 correlates with some RV echocardiographic parameters - combination of miRNAs (miR-20a-5p, miR-93-5p and miR-17-5p) => best predictor of RV function	35488248/[59]
**- miR-21**	N/A	- Sheep (5 mo) - PAB	- NRVM - phenylephrine	- mimic-21 - antagomir-21	In vivo: PAB vs. ctrl => ↓ RV function and basal stain RV PAB vs. ctrl basal RV => ↓ miR-21 expression PAB vs. ctrl apical RV => no changes in miR-21 expression In vitro - ↑ miR-21 alters mitosis and cytokinesis - ↓ miR-21 => ↓ phenylephrine induces hypertrophy. - ↑ miR-21 => ↑ phenylephrine induces hypertrophy.	33548242/[60]
**- RNA seq** **- qRT-PCR (miR-335-3p)**	N/A	Rats, male, Sprague Dawley, 6 weeks -MCT (60 mg/kg, i.p.) Mice, male, C57/BL6, 8 weeks -SuHX, 20 mg/kg s.c., 10% O_2_	H9c2	In vivo AntagomiR-335-5p In vitro AntagomiR-335-5p AngII	In vivo - MCT vs. ctrl rat RV => ↑ 74 miRNA and ↓ 77 miRNA. - MCT vs. ctrl rat RV => ↑ miR-335-5p - SuHX vs. ctrl mice RV => ↑ miR-335-5p - ↓ miR-335-5p => ↓ RV fibrosis and CM apoptosis, ↑ *calumenin* expression, and ↑ RV function in MCT rats In vitro - Ang II induces H9c2 hypertrophy => ↑ miR-335-5p - miR-335-5p binds calumenin - ↓ miR-335-5p => ↓ H9c2 hypertrophy, apoptosis, expression of cardiac stress markers (*ANP*), ↑ *calumenin* expression.	36246958/[61]
**miR-200c**	N/A	- Rats, Sprague Dawley, 3–4 weeks, male, 120–150 g - PAB	N/A	N/A	PAB vs. Sham RV => ↓ miR-200c	35384363/[62]
**Pulmonary stenosis and insufficiency**						
**- miR-1** **- miR-21**	N/A	- Rats, Wistar, females, 8 w - PAB, - Training (treadmill, 8 w).	N/A	N/A	- PAB vs. Sham RV => ↓ miR-21 - Training => ↑ miR21, ↓ miR-1 in RV	27994552/[32]
**- miR-21** **- miR-221** **- miR-222** **- miR-143**	RV infundibular muscle bundles: - 3 RVOTO (67% male, 2.9 y +/− 1.2, 67%) - 7 TOF (57% male, 0.3 y +/− 0.05), - 4 PS + PI (25% male, 3.4 y +/− 1.3) - 2 PS-RVF (100% male, 7.7 y +/− 4.1). Plasma: - 10 ctrl (40% male, 14.5 y +/− 3.6) - 7 PS + PI (50% male, 10.5 y +/− 4.5), - 9 PS-RVF (67% male, 8.7 y +/− 5.6)	- Mice, male, FVB, 12 w -PI + PS surgical model (pulmonary valves leaflet ligation and pulmonary artery banding)	N/A	N/A	Human: - PI + PS RVF vs. TOF and RVOTO => ↑ miR21 in RV - PI + PS vs. ctrl => ↑ miR-21 in blood - PI + PS-RVF vs. ctrl => ↓ miR21 in blood - RV miR-21 levels are associated with RV fibrosis in human and mice. - circulating miR-21 correlates with RV function in human and mice. In vivo: - PI + PS mices => ↑ RV miR-21, miR-221, miR-222. - PI + PS mices => ↓ blood miR-221, miR222. - ↑ blood miR-21 in early stage of PI+PS then ↓ miR-21 expression in late stages.	28469078/[63]
**Right ventricular failure**						
**- Microarray**	N/A	- Mice, FVB, male, 12 w - PAB	N/A	N/A	- PAB vs. sham => ↑ RV miR-199a-3p - PAB induced RVF => ↑ miR-208b miR-34, miR-1, ↓ miR-21 in RV. - ↑ 34a, 28, 148a, and 93 in RV but not in LV	22454450/[64]
**Systemic right ventricle**						
**- miR-423-5p**	Plasma - 41 SRV (65.9% male 29.2 y ± 3.6). - 10 ctrl (70% male, 30.3 y ±.4.6).	N/A	N/A	N/A	- SRV vs. Ctrl => no changed in miR-423-5p - No correlation between miR-423-5p and a clinical parameter.	22188991/[65]
**- Microarray**	Plasma: Discovery (microarray). - 5 TGA + SRV (60% male, 26.5 y ± 5.1) - 5 ctrl (60% male, 24.5 y ± 6.3). Validation (qRTPCR) - 26 TGA + SRV (65% male, 25.3 y ± 3.2) - 20 ctrl (55% male, 25.0 y ± 4.3).	N/A	N/A	N/A	- TGA + SRV vs. ctrl => ↑ miR-16, miR-106a, miR-144 *, miR-18a, miR-25, miR-451, miR-486-3p, miR-486-5p, miR-505 *, let-7e and miR-93 - miR-18a and miR-486-5p correlate with SRV isovolumic contraction.	24040857/[66]
**- Microarray**	Plasma: - 36 TGA + SRV (64% male, 36.3 y ± 12.3), - 35 ctrl (age sex matched)	N/A	N/A	N/A	- TGA + SRV => 106 miRNA with impaired expression. - miR-150-5p, miR-1255b-5p, miR-423-3p, and miR-183-3p associated impaired RV. - miR-183-3p => independent predictors of worsening HF	34568463/[67]
**Tetralogy of Fallot**						
**- RNAseq**	RV: - 22 TOF (54% male, 0–3 y), - 4 ctrl (50% Male, 14–25 y)	N/A	N/A	N/A	- TOF vs. ctrl: 172 differentially expressed miR, - Disease-related miRNA-mRNA pairs include miR-1 and miR-133, which are essential to cardiac development and function by regulating *KCNJ2, FBN2, SLC38A3* and *TNNI1*	31836860/[68]
**- Microarray**	RVOT: Discovery (microarray) - 10 TOF, - 6 ctrl Validation (qRT-PCR) - 26 TOF, - 15 ctrl	N/A	- hCMPCs	- Mimic-miR-940 - Inhibitors-miR-940	Humans: - TOF vs. ctrl => 75 differentially expressed miRNA - TOF vs. ctrl => ↓ miR-940 expression In vitro: - ↓ miRNA-940 expression => ↑ hCMPCs proliferation and ↓ migration. - ↑ miRNA-940 expression => ↓ hCMPCs proliferation - miRNA-940 targets *JARID2*.	24889693/[69]
**- miR-421**	- RV myocardium: Discovery (microarray) - 16 TOF (69% male, 276 d (98–510). - 8 ctrl (37% male, 142 d (28–382). Validation (qRT-PCR) - 8 TOF (50% male, 292 d (167–425), 4M/4F)	N/A	- HRVPCC	- Plasmid-miR-421 - Inhibitors-miR-421	Humans - TOF vs. ctrl => ↑ miR-421 – In Vitro - miR-421 targets *SOX4*	25257024/[70]
**- Microarray**	RVOT Discovery (microarray) - 5 non-syndromic TOF (60% male, 28.6 m +/− 4.7) - 3 ctrl (33% male, 10.7 m +/− 8.1). Validation (qRT-PCR) - 26 TOF (69% male, 11.7 m +/− 8.7) - 6 ctrl (33% male, 21.2 m +/− 14.4).	N/A	- P19 - PEMC (E12.5)	- Lentivirus miR-222 (overexpression) - Lentivirus miR-424/424* (overexpression)	Humans: - TOF vs. ctrl => ↑ miR-146b-5p, miR-155, miR-19a, miR-222, miR-424, miR337-5p, miR363, miR-130b, miR-154, miR-708, miR-181c, miR-424*, miR-181d, miR-192, miR-660 - TOF vs. ctrl => ↓ miR-29c, miR-720, miR-181a*. In vitro: - miR-424/424* ↑ PEMC proliferation and ↓ migration - miR-424/424* ↓ *HAS2* and *NF1* expression. - miR-222 ↑ PEMC proliferation - miR-222 ↓ P19 cardiomyogenic differentiation.	24140236/[71]
**- Microarray**	Plasma: - 4 TOF with normal RV (100% male 29.0 y (22.4–36.6)) - 11 TOF with mild/moderate RV enlargement (36% male, 35.8 y (26.3–40.7)) - 5 TOF with severe RV enlargement (40% male; 46.5 y (42.7–51.2))	N/A	N/A	N/A	- normal RV size vs. mild-moderate and severe RV enlargement => ↓ 267 miRNA and ↑ 66 miRNA. - miRNA 28-3p, 433-3p, and 371b-3p => associated with ↑ RV size and ↓ RV systolic function. - Dysregulated miRNAs => cell cycle pathways, extracellular matrix proteins and fatty acid synthesis pathways. - ↓ HIF 1α signaling (predicted) - ↑ p53 signaling (predicted)	33175850/[72]
**- Microarray**	Plasma - 15 ctrl (30.2 ± 10.8 years), - 3 TOF + RVF (33.0 y +/− 13.9) - 34 TOF no RVF (29.9 y +/− 10.7)	N/A	N/A	N/A	All TOF vs. ctrl - 49 miRNA impaired - ↓ miR181d-5p, miR-142-5p, miR-206 (PCR validation) - ↑ miR-625-5p (PCR validation) TOF + RVF vs. ctrl - 58 miRNA impaired - ↓ miR-181d-5p, miR-206 (PCR validation) - ↑ miR-1233-3p, miR-183-5p, miR-421, miR-625-5p (PCR validation) TOF no RVF vs. ctrl - 77 miRNA impaired - ↓ miR-421, miR-1233-3p, miR-625-5p miR-421 and miR-1233-3p correlate with RV function (ejection fraction, end diastolic and end systolic volume).	28693530/[73]
**- Microarray**	Plasma: Discovery (micro array) - 8 TOF (62% male, 21.5 y +/− 6), - 8 ctrl (62% male, 19.7 y +/− 4.2). Validation 1 (qRT-PCR) - 49 TOF (51% male, 23.0 y +/− 6.7), - 30 ctrl (50% male, 22.7 y +/− 5.8). Validation 2 (qRT-PCR) - 55 TOF (49% male, 23.3 y +/− 6.5), - 40 ctrl (62.5% male, 23.4 y +/− 5.3). Validation 3 (qRT-PCR) - 104 TOF (50% male, 23.2 y +/− 6.6), - 70 ctrl (57% male, 23.2 y +/− 5.5)	N/A	N/A	N/A	TOF vs. ctrl => ↑ circulating miR-99b TOF vs. ctrl => ↓ circulating miR-766	28664568/[74]
**MicroArray**	RV myocardium: - 16 TOF (31% female, 276 d (98–510), - 8 ctrl (62% female, 142 d (28–382)), - 8 TOF (validation, 50% female, 292 d (167–425)), -3 fetal samples.	N/A	N/A	N/A	- TOF vs. ctrl => 61 miRNAs differentially expressed miRNA (32 upregulated, 29 downregulated)	22528145/[75]
**Microarray**	RV - 4 TOF (50% male, 2.7 y +/− 1.5) - 4 ctrl (10.7 y +/− 0.3)	N/A	N/A	N/A	TOF vs. ctrl => ↑ 16 miRNAs↓ 31 miRNAs	22882842/[76]
**miR-21**	Blood - 60 TOF (47% male, 15.0 y (9, 22))	N/A	N/A	N/A	- miR-21 levels did not correlate with RV functional parameters	35487317/[77]
**Micro-array**	RV - 14 TOF (50% male, 7 m +/− 2.7)	N/A	N/A	N/A	Female vs. male TOF => 41 differentially expressed miRNA including miR1/133	30150777/[78]

ACM: Arrhythmogenic Cardiomyopathy; ARVC: Arrhythmogenic right ventricular cardiomyopathy; CABG; coronary artery bypass graft; CAD; coronary artery disease; CAV: cardiac allograph vasculopathy; CHD: Congenital heart diseases; CM: cardiomyocytes; CRV: compensated RV; CTEPH: Chronic thromboembolic pulmonary hypertension; Ctrl: control; DCM: dilated cardiomyopathy; DRVF: dog RV fibroblast; EC: endothelial cells; EMCM; embryonic mouse cardiomyocytes; HF: heart failure; HLHS: Hypoplastic left heart syndrome; HX: hypoxia; i.p: intraperitoneal; hCMPC: human cardiomyocyte progenitor cells; HRVPCC: human RV primary cell culture; IPAH: idiopathic pulmonary arterial hypertension; IPF: idiopathic pulmonary fibrosis; IVT: interventricular tachycardia; LV: left ventricle; LVAD: left ventricular assist device; LVRR: left ventricular reverse remodeling; MCT: monocrotaline; NRC: neonatal rat cardiomyocyte; NRVM: neonatal rat ventricle myocyte; PAB: pulmonary artery banding; PAH: pulmonary arterial hypertension; PEMC: primary mouse embryonic cardiomyocyte; PH: pulmonary hypertension; PI: pulmonary insufficiency; PS: pulmonary stenosis; RCF: rat cardio-fibroblast; RV: right ventricle; RVF: right ventricular failure; RLVC: rat LV cardiomyocyte; RRVC: Rat RV cardiomyocyte; RVD/SD: RV dilatation/systolic dysfunction; RVH: RV hypertrophy; RVOT: right ventricular outflow tract; RVOTO: RV outflow tract obstruction; S.C.: subcutaneous; SuHX: sugen + Hypoxia; VA: ventricular arrhythmia; SRV: systemic RV; TGA: transposition great arteries; TOF: tetralogy of Fallot; VT: ventricular tachycardia;.

### 3.2. Arrhythmogenic Cardiomyopathy

ACM is a genetic disease [79,80,81,82,83] characterized by a progressive fibro-fatty substitution of the ventricular myocardium, which is particularly pronounced in the RV, associated with life-threatening ventricular arrhythmias and, ultimately, heart failure [84,85]. Traditionally, ACM was known as arrhythmogenic right ventricular cardiomyopathy (ARVC), stemming from the initial belief that the condition was confined exclusively to the RV chamber of the heart. ACM diagnosis is challenging and is often achieved after disease onset or postmortem [86]. The quantification of circulating miRNA represents an attractive approach to uncover novel biomarkers. For example, two independent studies reported increased miR-185-5p levels in the blood of ACM patients compared to control patients (the sample consisted of healthy controls, patients with idiopathic ventricular tachycardia, and patients with borderline or possible ACM) [19,21]. Interestingly, circulating miRNA allows for a granular classification of ACM and an increased miR-494 expression successfully discriminates ACM patients with recurrent ventricular arrhythmia post radiofrequency catheter ablation from patients without recurrent ventricular arrhythmia [21]. Similarly, Sommariva and colleagues reported that decreased levels of circulating miR-320a successfully discriminate ACM from idiopathic ventricular tachycardia and healthy patients [15]. Although miR-185-5p, miR-494, and miR-320a might represent a potent biomarker for ACM, whether their expression is also affected in cardiac and RV tissue remains uncertain. Comparing both cardiac tissue and blood, Bueno Marinas et al. reported that six miRNAs (miR-122-5p, miR-133a-3p, miR-133b, miR-142-3p, miR-182-5p, and miR-183-5p) are differentially expressed in the blood and RV of ACM patients compared to healthy controls [17]. Interestingly, miR-122-5p (as well as miR-1-3p, miR-21-5p, miR-206, and miR-3679-5p) was also differentially expressed in pericardial fluid from ACM patients, compared to patients with post-infarct ventricular tachycardia [20]. Increased miR-21-5p and decreased miR-135b expression observed in ACM in the RV correlate with myocardial fibrosis and adipose tissue deposition [17]. Nonetheless, whether targeting miR-21-5p and miR-135b improve ACM and have a therapeutic effect remains unexplored. Mazurek and colleagues demonstrated that forced miR-130a expression results in ACM phenotypes (leading to RV hypertrophy, arrhythmogenic dysfunction associated with RV fibrosis, lipid accumulation, and cardiomyocyte death) in mice [16]. Functionally, they demonstrated that miR-130a mediated the translational repression of *Desmocollin2*, an important protein for cell adhesion that is dysregulated in human ACM (OMIM 610472). Interestingly, treatments or therapeutic interventions can also affect miRNA expression profiles. Thus, compared to non-paced children, blood from chronically paced children exhibit an increased and decreased expression of 296 and 192 miRNA, respectively [22]. In conclusion, the study of miRNA is mainly descriptive and restrained to the biomarkers field in ACM.

### 3.3. Pulmonary Hypertension

Pulmonary hypertension (PH) is hemodynamically defined by an increased mean pulmonary arterial pressure above 20 mmHg, which imposes a substantial hemodynamic stress load on the RV [87]. As a consequence of this inescapable afterload rise, patients exhibit a progressive decline in cardiac function, which culminates in RV failure and premature death. The current classification system divides PH into five groups, including pulmonary arterial hypertension (PAH or group 1 PH), PH due to left heart disease (PH-LHD or group 2 PH), PH due to lung disease and/or hypoxia (or group 3 PH), chronic thromboembolic PH (CTEPH or group 4 PH), and PH with unclear and/or multifactorial mechanisms. Regardless of the group, RV function is one of the most important predictors of both morbidity and mortality in PH. In the past 5 years, tremendous efforts have been undertaken to understand the molecular mechanisms underlying RV failure in PAH. In this context, epigenetic modifications, including miRNA, have been widely studied.

The first step was to use a large screening array (e.g., a microarray) to characterize the miRNA signature associated with RV dysfunction in human PAH and preclinical models of the disease. Chouvarine and colleagues investigated the trans-RV and transpulmonary miRNA gradients and reported an increased expression of miR-29a-3p, miR-26a-5p, miR-590-5p, and miR-200c-3p in the PAH-superior vena cava, and a decrease in miR-99a-5p in the PAH-pulmonary artery, compared to the controls [50]. Moreover, they reported a transventricular miRNA gradient, comparing the pulmonary artery and the superior vena cava, with miR-193a-5p and miR-423-5p being regulated in opposite directions in PAH and control patients. In a preclinical setting, the RV from mice exposed to hypoxia exhibited the impaired expression of nine miRNAs (let-7e-5p, miR-29c-3p, miR-127-3p, miR-130a-3p, miR-146b-5p, miR-197-3p, miR-214-3p, miR-223-3p, and miR-451) [58]. Hypertrophic RV from the Sugen (VEGFR blocker) + hypoxia-induced PH rat model exhibited an increased expression of ten miRNAs (miR-21-5p, miR-31-5 and 3p, miR-140-5 and 3p, miR-208b-3p, miR-221-3p, miR-222-3p, miR-702-3p, and miR-1298) and a decrease in three miRNAs (miR-187-5p, miR-208a-3p, and miR-877) [58]. Reddy and colleagues characterized the miRNA landscape of a preclinical mouse model of RV failure induced by pressure overload (pulmonary artery banding surgery, PAB). Compensated RVs (early PAB) exhibited an increased expression of miR-199-3p, miR-199b*, and miR-223, and a decreased expression of eight miRNAs (let-7g, miR-139-5p, miR-143, miR-145, miR-151-5p, miR-181a, miR-26a, and miR-30a*), whilst decompensated RVs (late PAB) showed an increased expression of nine miRNAs (miR-127, miR-136, miR-199b, miR-208b, miR-221, miR-34b-5p, miR-34c, miR-379, and miR-503) and a decreased expression of nineteen miRNAs (miR-101a, miR-144, miR-185, miR-203, miR-208a, miR-218, miR-219, miR-29c*, miR-30b*, miR-30c-2*, miR-338-3p, miR-345-5p, miR-378, miR-451, miR-486, miR-582-5p, miR-669c, miR-709, and miR-805) [64]. miR-200c expression is also decreased in PAB rats’ RVs [62]. To further investigate the key microRNAs involved in RV dysfunction and failure, Drake and colleagues assessed microRNA expression in the RV of two PH or pressure overload rat models with compensated RV hypertrophy (PAB and chronic hypoxia) and two with decompensated RV (Sugen + hypoxia and PAB + copper diet) [57]. They reported a decreased expression of miR-133a, miR-139-3p, miR-21, and miR-34c in both models of decompensated RVs. Kameny and colleagues reported the impaired expression of 40 miRNAs in the RV from an ovine model of PH associated with congenital heart disease. Among them, a decreased level of miR-199b and an increased level of mir-29 were associated with an increased nuclear factor of activated T cells/calcineurin signaling and a decreased level of fibrosis, respectively [49].

Beyond the large-scale characterization of the miRNA landscape, miR-21 is the most studied miRNA in PH–RV dysfunction. In group 3PH patients, circulating miR-21 correlated with RV dysfunction (decreased tricuspid annular plane systolic excursion and tissue Doppler RV S′ wave) [51]. Moreover, an artificial increase in miR-21 induced cardiomyocyte hypertrophy and the expression of a cardiac stress marker (namely, brain natriuretic peptide) [60]. miR-21 was also increased in the RV, but not the LV, of a group 2 PH canine model [48]. Interestingly, Chang and colleagues reported dynamic changes in miR-21 expression in chronic heart disease-associated PAH patients (CHD–PAH) [38]. Compared to CHD–PAH without heart failure, CHD–PAH patients with heart failure had decreased circulating mir-21 levels, which are associated with increased heart failure hospitalization. In a CHD–PAH rat model, RV miR-21 expression was increased in early response to aorto-venous fistula, and then decreased. Similarly, cardiomyocyte exposed to shear stress exhibited an increased and decreased miR-21 expression at 3 h and 6 h post-treatment, respectively. Functionally, decreased miR-21 expression is associated with cardiomyocyte apoptosis, which can be reversed by forced miR-21 expression. This observation was confirmed by Liu and colleagues, who reported that decreased miR-21 expression in a decompensated RV from a preclinical model of PH was associated with increased cardiomyocyte apoptosis [39]. However, as observed in human plasma, whether the miR-21 expression is increased or decreased in the RV from preclinical models of PH or pressure overload remains conflictual. For example, the miR-21 expression was reported to be increased in RVs from female monocrotaline (MCT) rats [40] and male Sugen + hypoxia rats [31], but decreased in PAB female rats [32] and in decompensated (male Sugen + hypoxia rats and PAB + low copper diet) vs. compensated RVs (hypoxia and PAB) [57]. Moreover, miR-21 expression was decreased in the basal part of the RV and unchanged in the apex of PAB animals, suggesting a topologic regulation of miR-21 in this model [60]. Combined with the dynamic regulation of miR-21 [38], this topologic regulation of the miRNA might explain some discrepancies between the studies. miR-223 is another example of the complex regulation of miRNA in PAH-associated RV dysfunction. miR-223 was increased in MCT rat RVs [40] and decreased in hypoxia–PH and PAB mice [26,58]. Although an antagomir-223 treatment failed to decrease miR-223 expression and improve RV function in MCT rats [40], the genetic depletion of miR-223 decreased the RV function [26]. Moreover, artificial miR-223 over-expression improved RV function and decreased RV fibrosis, which was likely mediated by the lower expression of the *insulin-like growth factor-I receptor* (*IGF-IR*, a miR-223 target). Similarly, Levchenko and colleagues reported increased miR-197 and miR-146b levels in RVs from human PAH patients and Sugen + hypoxia rats, when compared to healthy donors and control animals [27]. In vitro, the authors demonstrated that miR-197 and miR-146b regulated metabolism, and that forced miR-197 and miR-146b expressions decreased the expression of factors that drive fatty acid oxidation (e.g., *CPT1B*, and *FABP4*). Interestingly, Chouvarine and colleagues observed increased mir-197 and decreased miR-146b expression in the RVs of mice exposed to hypoxia [58]. In vitro, they reported that hypoxia decreased miR-146b and induced *TRAF6*, *IL-6*, and *CCL2* (MCP-1) pro-inflammatory signalling. This controversial observation, finding a putative beneficial and deleterious role of miR-146b on RV function, highlights the complexity of miRNA regulation and contribution upon various stimuli.

Functionally, miRNA provided insight into the mechanisms regulating RV hypertrophy in PH. Connolly and colleagues reported that miR-424(322) expression was decreased in MCT rat RVs and likely contributed to the increased expression of the pro-hypertrophic growth factor insulin-like growth factor-1 [28]. However, Baptista and colleagues showed that increased levels of circulating miR-424(322), observed in PH patients, correlated with higher symptom severity and hemodynamics (e.g., miR-424(322) inversely correlated with cardiac output) [29]. In a subgroup of CHD–PAH, miR-424(322) was an independent marker of survival. Using a translational approach, the authors demonstrated that increased levels of circulating miR-424(322) were associated with RV hypertrophy in MCT rats. miR-214 expression was increased in RVs harvested from MCT rats, Sugen + hypoxia rats, and mice models of PH [30,33]. Moreover, in Sugen + hypoxia mice, the genetic depletion of miR-214 led to increased RV hypertrophy, likely mediated by *PTEN* overexpression [30]. *PTEN* is an important regulator of cardiac hypertrophy, which is also regulated by miR-495 in failing MCT rats’ RVs [34]. Decreased miR-495 expression was associated with increased *PTEN* and decreased cardiac stress markers (e.g., *NPPA*) in MCT RVs and cultured cardiomyocytes. Increased miR-335-5p was associated with a decreased expression of calumenin (miR-335-5p target) and pathologic RV hypertrophy in preclinical models of PH [35]. The inhibition of miR-335-5p expression decreases RV fibrosis, cardiomyocyte hypertrophy, and apoptosis, and improves RV function.

miRNAs are differentially expressed across cell types and tissues. For example, myomiRs are a class of miRNAs that are mainly expressed in muscles. Kmecova and colleagues studied the expression of four myomiRs (miR-1, miR-133a, miR-208, and miR-499) which are known to be enriched in cardiac tissue [33]. They reported a global decreased expression of those four cardio-myomiRs in RVs from MCT rats, compared to control animals [33]. Decreased miR-1 expression in MCT RVs was associated with increased TGF-β signalling, which might contribute to cardiac hypertrophy [36]. Intriguingly, miR-1 expression was unchanged in RVs from female PAB rats [32] but increased in hypertrophic RVs isolated from rats exposed to hypoxia [37]. Moreover, under hypoxic conditions, miR-1 silencing decreased RV hypertrophy, RV fibrosis, and fibroblast collagen production [37]. Mir-133a expression was also decreased in decompensated RVs (Sugen + hypoxia or PAB + low copper diet), when compared to animals with compensated RVs (hypoxia or PAB surgery) [57]. The expression of miR-208, and specifically miR-208a, was also decreased in RVs from Sugen + hypoxia PH rats [31]. Paulin and colleagues demonstrated that miR-208a expression in the RV negatively correlated with RV hypertrophy in MCT rats [53]. Moreover, they reported that decreased miR-208a activated the complex mediator of the transcription 13/nuclear receptor corepressor 1 axis, which in turn, promoted *Mef2* inhibition, likely contributing to cardiomyocyte hypertrophy and decreased angiogenesis.

AngiomiRs are a category of miRNA involved in the regulation of angiogenesis [88]. Among them, miR-126 is one of the most studied and important angiomiRs. Interestingly, a decreased mir-126 expression in PAH decompensated RVs correlates with capillary rarefaction and decreased RV function in both human patients and MCT rat models of PH [56]. This effect was likely mediated by the increased expression of *SPRED-1*, a downstream inhibitor of the VEGF/VEGFR2 angiogenic pathway. Conversely, artificial miR-126 overexpression decreased SPRED-1 expression, improved RV endothelial cells’ angiogenic capacity, and increased RV capillary density and function in MCT rats. Interestingly, miR-126 was also decreased in RVs from Sugen + hypoxia rats’ models of PH [57] and in failing RVs from a dog model of group 2PH [48]. Beyond PAH, decreased miR-126 RV levels were associated with post-transplantation complications, such as cardiac allograft vasculopathy [23]. MiR-17 is another angiomiR that was decreased in the blood of PH patients (compared to healthy controls) [55], which correlated with RV function [59]. However, miR-17 was increased in the RVs of female rats exposed to MCT [40].

FibromiR refers to a miRNA involved in fibrotic processes. Yi Tang and colleagues demonstrated that decreased miR-325 expression in MCT rats’ RVs negatively correlated with RV fibrosis [52]. Moreover, forced miR-325 expression in rat fibroblasts and MCT animals decreased collagen deposition and relieved RV fibrosis. Decreased miR-200b expression observed in MCT rats’ RVs was associated with the increased expression of the pro-fibrotic *protein kinase c alpha* [54]. Functionally, miR-200b overexpression decreased *protein kinase c alpha* expression in both the RVs and the lungs of MCT rats. Powers and colleagues reported the impaired expression of 26 miRNAs, including increased miR-21 and miR-221, in RV samples harvested from a dog model of group 2PH [48]. Hormonal (aldosterone) and mechanical (stretch) stress increased miR-21 and miR-221 expression in canine RV fibroblasts (but not in LV fibroblasts). Similarly, the inhibition of miR-21 and miR-221 decreased RV (but not LV) fibroblast proliferation.

### 3.4. Congenital Heart Diseases

Congenital heart diseases (CHDs) are the most common type of birth defect and the leading cause of inherited perinatal and infant mortality. CHD refers to structural anomalies of the heart and blood vessels that arise during cardiac development and represents a broad spectrum of malformations, including septal and valve defects, and lesions affecting the outflow tract and ventricle. CHDs include tetralogy of Fallot (TOF), pulmonary atresia, and pulmonary valve stenosis, among other diseases. Qian Yang and colleagues reported 20 differentially regulated miRNAs in RV outflow tract (RVOT) samples isolated from CHD patients, compared to controls [24]. Among them, increased miR-29b-3p was associated with reduced cardiomyocyte proliferation, cardiac malformation, and lethality in zebrafish.

-Tetralogy of Fallot

TOF is a complex congenital heart defect consisting of pulmonary stenosis, ventricular septal defect, overriding aorta, and RV hypertrophy. TOF is the most common cyanotic congenital heart defect and accounts for about 10% of all congenital heart malformations. Although the contribution of miRNA in TOF etiology remains poorly understood, several studies characterized the miRNA landscape associated with TOF development in RV myocardium, RV outflow tract RVOT, or plasma samples from human patients.

Grunert and colleagues reported that 172 miRNA (111 upregulated and 61 downregulated) were differentially regulated in TOF RVs [68]. They identified miR-1 and miR-133 as potential key miRNAs in disease etiology, notably by regulating *KCNJ2*, *FBN2*, *SLC38A3*, and *TNNI1* expression. Interestingly, miR-1/miR133 clusters might contribute to the regulation of the 41 differentially expressed miRNA between male and female TOF patients, and thus contribute to a potential miRNA sexual dimorphism in TOF [78]. Comparing TOF to control RVs, Zhang et al. observed 47 differentially expressed miRNA, including upregulation of three members of the miR-29 family (known to be involved in LV failure) [76,89]. Bittel and colleagues identified 61 differentially regulated miRNA (32 upregulated and 29 downregulated) in myocardial RV samples harvested from TOF and control patients [75]. They showed that increased miR-421 expression decreases *SOX4* expression, a key regulator of the Notch pathway involved in the cardiac outflow tract and TOF development [70]. A study, investigating the expression of miRNA in RV samples from patients with TOF, RVOT obstruction, combined pulmonary insufficiency (PI), and stenosis (PS), with and without end-stage RV failure, reported increased miR-21 expression in tissue samples from PI/PS patients, compared to TOF and RVOTO patients. Circulating miR-21 expression correlated with RV function and exhibited a dynamic regulation, as it increased in the plasma of PI/PS patients without RV failure and decreased in patients with RV failure. However, Dilorenzo and colleagues did not report any association between circulating miR-21 and RV function assessed by MRI [77]. Thus, the potential biomarker value of circulating miR-21 remains to be further investigated.

Using RVOT samples, Jin Zhang and colleagues identified 16 miRNAs (including miR-424/424* and miR-222), that were predicted to target genes involved in heart development and CHDs (e.g., *PTEN*, *HAND1/2*, *BMPR2*, *NFATC1*) [71]. In vitro, the forced expression of miR-424/424* reduced *HAS2* and *NF1* expression, promoted the proliferation but inhibited the migration of primary embryonic mouse cardiomyocytes, increased miR-222, and impaired the expression of 74 other miRNAs in RVOT from TOF patients [69]. Among them, increased miR-940 expression was associated with decreased *JARID2* expression, increased cardiomyocyte progenitor cell proliferation, and reduced migration.

The quantification of circulating miRNA is an attractive approach to identifying novel and potent biomarkers. Lai and colleagues reported decreased expressions of miR-766 and miR-99b in the plasma of TOF patients [74]. Abu-Halima and colleagues identified 49, 58, and 77 circulating miRNAs that were differentially expressed in, firstly, TOF patients vs. controls, secondly, TOF patients with right heart failure vs. controls, and, thirdly, TOF patients without symptomatic right heart failure vs. controls [73]. Similarly, Weldy and colleagues assessed the expression of circulating RNA in TOF patients with normal RV, mild to moderate RV dilatation, and severe RV dilatation [72]. The authors identified (and validated) increased miRNA 28-3p, 433-3p, and 371b-3p to be associated with increased RV size and decreased RV systolic function.

### 3.5. The Systemic Right Ventricle

The RV, which normally supports low-pressure pulmonary circulation, has to go through various adaptive mechanisms to sustain the systemic load when placed in the systemic position. Compared to normal condition, the systemic RV (SRV) represents a distinctly different model in terms of its anatomic spectrum, short- and long-term adaptation, and clinical phenotype. The common clinical scenarios where an SRV is encountered are as follows: the complete transposition of the great arteries (TGA) with previous atrial switch repair; double discordance or congenitally corrected TGA, operated or native; double-inlet RV; and hypoplastic left heart syndrome, which occurs in patients with aortic atresia or hypoplasia who require reconstruction of the aorta from the pulmonary artery (Norwood protocol).

Decreased miR-486 expression was also observed in RV samples from patients with hypoplastic left heart syndrome and in a surgical model of the disease induced by aortopulmonary vascular grafts [25]. In vitro, forced miR-486 expression decreased *FOXO1* and *SMAD* signalling, and increased *STAT1* expression and cardiomyocyte contractility. Interestingly, Lai and colleagues reported/validated the increased expressions of 11 miRNA, including miR-486, in blood from patients with TGA + SRV. Moreover, they observed that miR-486 and miR-18 negatively correlated with systemic ventricular contractility. An independent study identified 106 differentially expressed miRNA in plasma samples from TGA + SRV [67]. Four miRNAs, namely, miR-150-5p, miR-1255b-5p, miR-423-3p, and miR-183-3p, were associated with the clinical worsening of heart failure, with miR-183-3p being the only independent predictor of worsening heart failure in a multivariate model. Nonetheless, Tutarel and colleagues reported that miR-423-5p levels were unchanged in TGA + SRV patients, compared to healthy controls, and failed as a biomarker of heart failure [65].

### 3.6. Dilated Cardiomyopathy

Dilated cardiomyopathy (DCM) is characterized by the enlargement and dilation of one or both of the ventricles, along with impaired contractility, defined as an LV ejection fraction (LVEF) of less than 50% [90]. While often restrained to the LV, patients with DCM may also develop diastolic dysfunction and impaired RV function. In DCM patients, impaired RV function is associated with increased morbidity and lower survival [91].

Limited studies focused on the RV in DCM. Nonetheless, the impaired expression of miR-21 and miR-133 was described in most of them. Although Parikh et al. reported decreased miR-21 expression in DCM RVs (compared to controls) [41], Wang and colleagues observed an increase in miR-21 [43], and Rubis showed no differences in miR-21 levels [44]. Similarly, depending on the study, miR-133a expression was reported to be decreased [43,44] or unaffected [41] in RVs from DCM patients, compared to controls. Moreover, miR-133a expression was increased in DCM patients with RV dysfunction, compared to patients without [45]. The observed discrepancies in miR expression levels could potentially be attributed to variations in the selection and characterization of control groups, for which there is often limited information reported (e.g., healthy controls vs. non-DCM patients). In plasma, miR-21 levels correlated with RV morphology. Circulating Mir-133a was associated with RV morphology [42] and RV dysfunction [45], and was an independent predictor of mortality in DCM [44].

### 3.7. Right Ventricle Dysfunction Associated with Neonatal Hypoxia and Irradiation

Rats exposed to early hypoxia (2–20 days post-natal) exhibited increased inflammation and RV hypertrophy, which was associated with decreased miR-206 expression [47]. Interestingly, decreased miR-206 expression was also observed in the blood of children suffering from TOF [73] and patients with ARVC [20], suggesting that, beyond contributing to RV hypertrophy, miR-206 might be involved in several RV-related pathological processes. Radiation of the chest during cancer therapy is deleterious to the heart, mostly due to oxidative stress and inflammation-related injury. Indeed, RV and LV dysfunction and remodeling are common complications observed in human patients exposed to radiotherapy [92]. Similar to other types of RV dysfunction, RV from rats exposed to a single sub-lethal dose of irradiation exhibited an increase in miR-21 expression which was reversed by aspirin, atorvastatin, and sildenafil treatment [46], suggesting that this miRNA might be a critical determinant in RV function and homeostasis.

### 3.8. Long Non-Coding RNA

Non-coding RNA with a length of >200 nucleotides are classified as lncRNA. During the past decade, numerous studies have highlighted the importance of lncRNA in orchestrating heart development, cardiovascular cell signalling, and cardiac function [93]. Functionally, lncRNAs exert a plethora of functions, which make them key regulators of biological functions, but complicate their study. Indeed, depending on their localization and their specific interactions with DNA, RNA, and proteins, lncRNAs can modulate chromatin function, regulate the assembly and function of nuclear bodies, alter the stability and translation of cytoplasmic mRNAs, and interfere with signalling pathways. Tissue-specific and condition-specific expression patterns suggest that lncRNAs are potential biomarkers and provide a rationale to target them clinically. Studies investigating the role of lncRNAs in RV physiologic and pathologic development are summarized in Table 2.

lncRNAs are critical players in RV development. Touma and colleagues investigated the neonatal LV and RV and identified 21,916 known and 2033 novel lncRNAs. One hundred and ninety-six exhibited significant dynamic regulation along the maturation process, and eleven were specifically upregulated in the RV [94]. Keeping in mind that lncRNAs are poorly conserved across the species, the authors identified five lncRNAs with conserved orthologs in humans, including *UCP2-lncRNA* (NONMMUT062940), *n420212* (NONMMUT041263), *Fus*-lncRNA (NONMMUT063779), *Ppp1r1b*-lncRNA (NONMMUT011874), and *H19* (NONMMUT064276). *Heart And Neural Crest Derivatives Expressed 2* (*Hand2*) is a critical gene for the embryonic development of the heart and the differentiation of the outflow tract myocardium, RV, and right atria [95]. *Hand2* is expressed throughout the primary heart tube during early embryonic stages, with dominant expression in the RV and outflow tract as the heart tube loops. Consequently, *Hand2*-null (*Hand2^−/−^*) mice showed severe RV hypoplasia and growth delay from E9.5, with death by E10.5. Ritter and colleagues demonstrated that the lncRNA locus *Handsdown* (*Hdn*), active in the early heart cells, regulated *Hand2* expression and was essential for early mouse development [96]. Indeed, the transcriptional activity of the *Hdn* locus, independent of its RNA, suppressed its neighboring gene *Hand2*. Consequently, *Hdn*-heterozygous adult mice exhibited increased *Hand2* expression and RV wall hyperplasia. Interestingly, Andersson and colleagues demonstrated that another lncRNA, namely, *upperhand* (*Uph*), is required to maintain *Hand2* transcription [97]. They reported that *Uph*-heterozygous adult mice exhibited decreased *Hand2* expression and RV wall hypoplasia. Those data suggest a tight epigenetic regulation of *Hand2* in the embryogenic development of the heart and RV.

The RV myocardium of an acute rat model of PH right heart failure exhibited an impaired expression of 169 lncRNA [98]. Consistent with the potential role of lncRNA in PAH RV dysfunction, Omura and colleagues reported that increased expression of the lncRNA *H19* correlated with RV fibrosis and cardiomyocyte hypertrophy in PAH patients and two preclinical models of RV failure [99]. Notably, *H19* is known to serve as the primary precursor of miR-675, which is also increased in decompensated RV. Functionally, they reported that targeting *H19* (siRNA) improved RV function and decreased RV fibrosis and hypertrophy in two preclinical models of RV failure. In vitro, silencing *H19* decreased cardiomyocyte hypertrophy and the expression of cardiac stress genes, while *H19* overexpression had the opposite effect. Moreover, the authors reported that increased circulating *H19* in the blood of PAH patients with decompensated RVs was an independent predictor of survival in PAH. This study demonstrated the therapeutic potential and the biomarker value of lncRNA in PAH–RV failure.

Focusing on lncRNA, several studies dissected RNA-sequencing data to describe the changes in the non-coding RNA landscape associated with RV dysfunction across various pathological conditions. RV samples from ischemic and non-ischemic RV failure expressed over 10,831 lncRNA [100]. Di-Salvo and colleagues demonstrated that RV myocardium harvested from human patients with ischemic and non-ischemic RV failure exhibited impaired expressions of 105 lncRNAs [101]. Similarly, RV myocardium from patients with cardiomyopathy and tricuspid regurgitation exhibited impaired expressions of 163 lncRNAs and 201 miRs [102]. Another descriptive study demonstrated that hypertrophic RV (from TOF/pulmonary stenosis patients) showed impaired expressions of twenty-five lncRNA and eight miRs [103]. Wang and colleagues observed that increased expression of the lncRNA *HA117* in the RVOT of TOF patients was associated with the severity of the disease and might lead to adverse short-term outcomes in TOF patients [104]. Beyond disease development and progression, Yaping Shan and colleagues reported that the lncRNA profile was affected by surgical procedures [105]. They observed that the implantation of expanded polytetrafluoroethylene’s (a biomaterial commonly used in cardiovascular surgery) into RVOT was associated with the impaired expression of 246 lncRNAs, likely contributing to RV inflammation and fibrosis in animals. Although descriptive, those studies suggest a potential role for lncRNA in RV dysfunction.

**Table 2 cells-12-02693-t002:** Long non coding RNA and right ventricle.

Epigenetic Modification	Human	In Vivo	In Vitro	Therapeutic Intervention	Outcome	PMID/Ref
**Development/Embryogenesis**						
**RNA seq**	RVOT samples: - 4 TOF (2 and 24 M) - 3 VSD (2 and 5 Y)	Mice, C57B/6, male postnatal D0 (before the ductal closure), D3, and D7	N/A	N/A	- In vivo, RV vs. LV => ↑expression of 11 lncRNA. - In human, lncRNA orthologs correlate with ↓ expression of their putative mRNA targets.	27591185/[94]
**LncRNA HDN**	N/A	C57BL/6 mice depleted for Hdn (*Hdn^+/em5Phg^*)	N/A	N/A	- lncRNA Hdn haploinsufficient for embryonic development. - Hdn RNA transcript dispensable for cardiac gene regulation. - Hdn regulates the cis-located *Hand2* - *Hdn^+/em5Phg^* adult mice => exhibit RV hyperplasia	31422919/[96]
**LncRNA Uph**	N/A	C57BL/6 mice lacking Uph (Uph^+/−^)	N/A	N/A	- *Uph^+/−^* => ↓ Hand2 expression, - *Uph^+/−^* => RV hypoplasia and embryonic lethality in mice. - Hand2 regulated by super-enhancer H3K27ac - *Hand2 ^+/−^* => RV hypoplasia and embryonic lethality. - Uph–Hand2 regulatory partnership can establish a permissive chromatin environment.	27783597/[97]
**Heart Failure**						
**RNA seq**	22 explanted human HF hearts: - 11 non ischemic RVF (36% female, 49 +/− 14 y), - 11 ischemic RVF (18% female, 54 +/− 9 y), control 5 unused donor human hearts (no info)	N/A	N/A	N/A	- RVF vs. ctrl => 105 differentially expressed lncRNAs,	25992278/[101]
**RNA seq**	RV myocardium - 22 RVF (84% female, 51 y +/− 11) - 5 ctrl	N/A	N/A	N/A	- Ctrl RV express 10,732 lncRNA - RVF express 10,831 lncRNA	25725476/[100]
**Pulmonary hypertension**						
**RNA seq**	N/A	- Rats, Male, Sprague-Dawley, 250–300 g - MCT (IP, 60 mg/kg, 28 D) - LPS, (IP, 1 mg/kg, serotype O55:B5, 2 h)	N/A	N/A	- 169 lncRNAs were differently expressed in PH rats with acute RV failure (MCT + LPS)	30119177/[98]
**- LncRNA H19,** **- miR-675**	RV samples: - 18 ctrl (55% male, 47 +/− 2.9 y) - 15 CRV (26% male, 31 +/− 12.2 y) - 11DRV (18% male, 62 +/− 11.5 y). Blood discovery cohort - 57 ctrl (35% male, 46 +/− 19 y) - 52 IPAH (40% male, 61 +/− 16 y), - 21 CTD PAH (24% male, 69 +/− 8 y) Blood validation cohort - 54 ctrl (32% male, 48 +/− 18 y) - 75 IPAH (73% male, 52 +/− 16 y) - 31 CTD-PAH (24% male, 66 +/− 9 y).	Rats, Male, Sprague-Dawley - MCT (SC, 60 mg/kg) - PAB	- H9c2 - NRVM	- H19 sirna, - adv-H19 - mimic 675	Human: - ↑ *H19* and mir 675 in PAH DRV. - H19 correlates with RV fibrosis and CM hypertrophy. - ↑ H19 in blood of PAH DRV patients. - Circulating H19 is an independent predictor of survival. In vivo: - targeting H19 improves RV function, ↓ RV fibrosis in MCT and PAB rats. In vitro: - si H19 => ↓ CM hypertrophy and expression of cardiac stress genes - adv H19 and mimic-675 has the opposite effect.	32698630/[99]
**Surgery**						
**RNA seq**	N/A	Rabbit, Female, new. Zealand, 3 month, 2.5–3.5 kg) - ePTFE implantation in RVOT. - Sham (suture) implantation.	N/A	N/A	ePTFE vs. Sham RV => ↑ 110 lncRNA, (potentially involved in the regulation of inflammation) ↓ 136 lncRNA (potentially involved in cardiac remodeling and metabolism)	34239925/[105]
**Tetralogy of Fallot**						
**HA117**	- RVOT tissues from 84 TOF (55% male, 13 (9,24.5) month)	N/A	N/A	N/A	- ↑ lncRNA HA117 is a risk factor for unfavorable McGoon ratio, Nakata index and LVEDVI in TOF. - ↑ lncRNA HA117 might lead to adverse short-term outcomes in TOF patients.	30258948/[104]
**RNA seq**	RV or RVOT tissue: - 19 TOF/PS (37% female, 5.4 +/− 1.3 m); 8 VSD (control, 63% female, 6.1 +/− 3.5 m)	N/A	N/A	N/A	TOF/PS vs. VSD => 8 miR, 25 lncRNA are differentially expressed	33786422/[58]
**Tricuspid regurgitation**						
**RNA seq**	RV myocardium: - 9 patients with TR cardiomyopathy (no info). - 9 Ctrl from the GTEx project (no info)	N/A	N/A	N/A	TR vs. Ctrl => 648 mRNAs, 201 miRs, 163 lncRNAs are differentially expressed	34603374/Tian C et al. 2021 [103]

Adv: adenovirus; CM: cardiomyocytes; CRV: compensated right ventricle; CTD PAH: connective tissue disease PAH; Ctrl: control; DRV: decompensated right ventricle; ePTFE: Expanded polytetrafluoroethylene; Hdn: Handsdown; HF: heart failure; IP: intraperitoneal; IPAH: idiopathic pulmonary arterial hypertension; LncRNA: long-noncoding RNA; LVEDI: left ventricle end diastolic volume Index; LPS: lipopolysaccharides; MCT: monocrotaline; NRVM: neonatal rat ventricular myocyte; PAB: pulmonary artery banding; pulmonary; PH: pulmonary hypertension; PS: pulmonary stenosis; RV: right ventricle; RVF: right ventricular failure; RVOT: right ventricular outflow tract; TOF: tetralogy of Fallot; TR: tricuspid regurgitation; Uph: upperhand; VSD: ventricular septal defect.

### 3.9. Other Non-Coding RNA

Beyond miRNA or lncRNA, the impaired expressions of other classes of non-coding RNA, namely circular RNAs (circRNAs) and small nucleolar RNAs (SnoRNAs), have been associated with RV dysfunction (Table 3). CircRNAs are single-stranded, covalently closed RNA molecules that are ubiquitous across species. CircRNAs exert biological functions by acting as transcriptional regulators, miRNA sponges, and protein templates. Circulating levels of CircRNAs have been proposed as potential biomarkers of RV function. For example, increased *circRNA_0068481* and decreased circulating expression of its “sponged miRNA” (miR-646 and miR-570) predicted RV hypertrophy in PH patients [35]. SnoRNAs are non-coding RNAs, found in the nucleolus, that primarily guide nucleotide modifications of rRNA (ribosomal RNA). Hypertrophic RV myocardium from TOF patients exhibited impaired expressions of 135 snoRNA [75] and 3 circRNA, namely, ENSG00000171517:*LPAR3*, ENSG00000155657:*TTN*, and ENSG00000127914:*AKAP9:chr7* [103].

**Table 3 cells-12-02693-t003:** Circular RNA, Small nucleolar RNA and right ventricle.

Epigenetic Modification	Human	In Vivo	In Vitro	Therapeutic Intervention	Outcome	PMID/Ref
**- circRNA_0068481** **- mir-646** **- miR-570** **- miR-885**	Blood - ctrl (92.3% female, 54.3 y +/− 5.9) - PAH no RVH (98% female, 45.5 y +/− 4.8) - PAH + RVH (100% female, 46.1 y +/− 3.7)	N/A	-AC-16	- CircRNA_0068481 siRNA	In human - PAH + RVH VS PAH no RVH and ctrl => ↓ *circRNA_0068481*, ↑ miR-646 and miR-570 - circRNA_0068481, miR-646 and miR-570 predict RVH In vitro - mir-646, miR-570, miR-885 bind circRNA_0068481 - ↓ *circRNA_0068481* => ↑ mir-646, miR-570, miR-885 expression	33710774/Guo HM et al. 2021 [35]
**RNA seq**	RV or RVOT tissue: - 19 TOF/PS (37% female, 5.4 +/− 1.3 m); 8 VSD (control, 63% female, 6.1 +/− 3.5 m)	N/A	N/A	N/A	TOF/PS vs. VSD => 3 CircRNAs, are differentially expressed (ENSG00000171517:*LPAR3*, ENSG00000155657:*TTN*, and ENSG00000127914:*AKAP9:chr7*)	33786422/[103]
**microArray**	RV myocardium: - 16 TOF (31% female, 276 d (98–510), - 8 ctrl (62% female, 142 d (28–382), - 8 TOF (validation, 50% female, 292 d (167–425), - 3 fetal samples.	N/A	N/A	N/A	- TOF vs. ctrl => 61 miRNAs and 135 snoRNAs differentially expressed - 44 genes had significant negative correlation with 33 miRNAs, - Potential link between levels of snoRNA, spliceosomal function, and heart development.	22528145/[12]

CircRNAs: Circular RNA; Ctrl: control; PAH: pulmonary arterial hypertension; PS: pulmonary stenosis; RV: right ventricle; RVH: right ventricular hypertrophy; SnoRNAs: small nucleolar RNA; TOF: tetralogy of Fallot; VSD: ventricular septal defect.

### 3.10. Histone Modifications

Histones directly interact with DNA and are the principal component of chromatin. Post-transcriptional modification of histones (including histone acetylation, methylation, phosphorylation, ubiquitination, etc.) is one of the main epigenetic mechanisms regulating chromatin compaction and subsequent gene expression. Histone modifications are tightly regulated processes. For example, histone acetylation status is regulated by two groups of enzymes exerting opposite effects; the histone acetyltransferases (HATs) and the histone deacetylases (HDACs). Similarly, histone methylation is regulated by enzymes which add (histone methyl transferase, HMT) or remove (histone demethylase) methyl groups from the histone. Impaired regulation of histone modifications results in aberrant gene expression and pathologic conditions, including LV failure [106]. Therefore, the development of pharmacological inhibitors of enzyme-regulating histone modifications (e.g., HDAC, HAT, and HMT) opens the doors to novel epigenetic-based therapeutic approaches. Studies focusing on the role of histone modification in RV physiologic and pathologic development are summarized in Table 4.

Histone modifications are also critical for regulating embryonic gene expression and developing the fetal heart, including the formation of the RV. Indeed, homozygous depletion of the HMT *SET-MYND-domain 1* (*Smyd1*) in cardiomyocytes and the outflow tract resulted in a truncated RV and outflow tract and embryonic lethality at E9.5 in mice [107]. Functionally, *Smyd1* depletion impaired the expansion and proliferation of the second heart field, which is involved in the differentiation of the RV and outflow tract. Similarly, homozygous depletion of the Hdac *m-BOP* gene resulted in decreased *hand2* expression, resulting in RV hypoplasia and ventricular maturation defects [108]. Genetically engineered mice lacking *Hdac2* exhibited abnormalities of the right chamber, with an almost complete loss of the RV lumen [109]. Beyond the expression, the cellular localization of HDAC is critical for RV development. For example, mice lacking apelin receptor *APJ* had greater endocardial *Hdac4* and *Hdac5* nuclear localization and reduced expression of *MEF2* and its transcriptional target *Krüppel-like factor 2*, resulting in poorly looped hearts and an aberrant RV [110]. In adult mice, enhanced recruitment of the histone acetyltransferase p300 led to increased acetylation of H3K9/14 and H4, as well as demethylation of H3K4. This epigenetic modification was associated with heightened expression of cardiac stress genes, including atrial and brain natriuretic peptides, predominantly in the left ventricle (LV) as compared to the right ventricle (RV). Interestingly, Notch depletion in mouse cardiomyocytes isolated from the LV resulted in increased H3kme3 marks on the *Hrt2* promoter, while Notch depletion had no effect on mouse cardiomyocytes isolated from the RV [111]. Taken together, those observations suggest that post-natal histone modification contributed to distinct gene expression patterns in the healthy LV and RV (likely reflecting the distinct afterload and wall stress applied to both cavities) [112].

HDACs are clustered into five classes, namely, HDAC class I (includes HDAC 1, 2, 3, and 8), HDAC class IIA (HDACs 4, 5, 7, and 9), HDAC class IIB (HDACs 6 and 10), HDAC class III (HDACs called sirtuins), and HDAC class IV (HDAC 11). RVs isolated from the PH rat models (SUHX, and MCT) and pressure overload (PAB) model exhibited increased activity of HDAC classes I, IIa, and IIb, while the RVs from animals exposed to hypoxia alone showed only increased activity of HDAC class IIb [113,114]. However, the expression of *Hdac 1*, *2*, and *3* (Class-I HDAC) was increased in RVs from rats exposed to hypoxia [115], and *Hdac 1* was increased in hypoxic calves’ RV tissues [116]. Consequently, the HDAC class I, II, and IV inhibitor, vorinostat, improved RV function in hypoxic calves [116]. The HDAC class I inhibitor sodium valproate decreased RV hypertrophy and fibrosis in the MCT and PAB rat models [117], and GCD0103, which selectively inhibited Hdac 1, 2, and 3 (HDAC class I) decreased the expression of cardiac stress genes, inhibited the proapoptotic caspase activity, and reduced the proinflammatory protein expression in the RV of hypoxic rats [115]. Conversely, in PAB rats, the HDAC class I and II inhibitor trichostatin-A failed to prevent RV hypertrophy, increased RV fibrosis, decreased RV capillary density, and increased cardiomyocyte apoptosis [118].

Increased expression of the Hdac *GCN5* was observed in RV samples from ACM patients and associated with lipid accumulation and reactive oxygen species (ROS) production [119]. Impaired post-translational histone modification mechanisms were associated with congenital heart diseases affecting the RV. For example, RV samples from children with single ventricle heart disease exhibited increased HDAC Class I, IIa, and IIb catalytic activity and protein expression [120]. RVOT from TOF children showed decreased mRNA and protein expression of the histone chaperone *HIRA* (which facilitates the activity of HDACs) [121].

**Table 4 cells-12-02693-t004:** Histone modification and right ventricle.

Epigenetic Modification	Human	In Vivo	In Vitro	Therapeutic Intervention	Outcome	PMID/Ref
**Embryogenesis**						
**HDAC4 and HDAC5**	N/A	C57BL/6 mice depleted for Apelin receptor *APJ^−/−^*	N/A	N/A	- HDAC4, and HDAC5, activates MEF2. - *APJ^−/−^* mice ↓ nuclear HDAC4, ↓ HDAC5, ↓ *MEF2* expression - *APJ^−/−^* mice => poorly looped hearts, aberrant RV, defective atrioventricular cushion formation	23603510/[110]
**HMT Smyd1**	N/A	C57BL/6 mice depleted for Smyd-1 (*Smyd^−/−^*)	N/A	N/A	- OFT and CM *Smyd^−\−^* => embryonic lethality at E9.5, - *Smyd^−\−^* => truncation of the OFT and RV - *Smyd^−\−^* =>impaired expansion and proliferation of the second heart field (SHF).	25803368/[107]
**HDAC mBOP**	N/A	C57BL/6 mice depleted for mBOP (*BOP^−/−^*)	N/A	N/A	*- BOP^−/−^* => single ventricle lack of RV marker (e.g. HAND2). - BOP is necessary for maturation of CM and morphogenesis of the RV.	11923873/[108]
**HDAC1** **HDAC2**	N/A	C57BL/6 Depleted for HDAC1 and/or HDAC2	N/A	N/A	- *HDAC2*^−/−^ => perinatal death => obliteration RV lumen	17639084/[109]
**Healthy RV**						
**- H3/H4 acetylation** **- H3 dimethylation** **- HAT P300**	N/A	Mice, Swiss Webster, 8 W	N/A	N/A	- RV vs. LV => ↓ BNP and ANP, and a-mhc expression. - RV vs. LV => ↓H3k4m2, ↓acetyl-H3k9/14, ↓acetyl H4 marks, and ↓ p300 recruitment on BNP and ANP genes - RV vs. LV => ↓ h3k4m2 on *a-mhc* and *b-mhc*	20090419/[112]
**Pulmonary hypertension**						
**HDAC Class I, IIa, IIb**	N/A	- Rats, Male, Sprague-Dawley, 250–280 g - Hypoxia (10% O_2_, 3 W) - SU5416 (20 mg/kg) + Hypoxia (10% O_2_, 3 W)	- NRVMs - ARVMs - ARVFs	N/A	In vivo - ↑ class I, IIa, IIb HDAC activity in Sug-Hx RV - ↑ class IIb HDAC activity in Hx RV - ↑ HDAC6 expression in Hx RV In vitro - ↓ HDAC-I in NRVM exposed to PE, NE, PGF2a, ET1 - ↑ HDAC-I activity in ARVM exposed to PE, NE, PGF2a, ET1, FBS, IL1b - ↓ HDAC-IIa activity in NRVM exposed to NE and ET1 -↑ HDAC-IIa activity in ARVM exposed to PE, PGF2a, FBS - ↑ HDAC-IIb activity in NRVM exposed to PE, NE, PGF2a, ET1 - ↑ HDAC-IIb activity in ARVM exposed to PE, NE, PGF2a, ET1, FBS, IL1b - ↑ HDAC-6 activity in NRVM and ARVM exposed to PE	21539845/[113]
**HDAC Class I**	N/A	- Rats, Male, Sprague-Dawley, 250–280 g - Hypoxia (10% O_2_, 3 W)	N/A	- Class I HDACs 1, 2, and 3 inhibitor MGCD0103 (daily IP, 10 mg/kg/days, 3 weeks). - Class I HDAC- inhibitor MS-275 (daily IP 3 mg/kg/days, 3 weeks).	- ↑ HDAC 1, 2, 3 in Hx RV. - MGCD0103 ↓ RVH, BNP expression, CM apoptosis, inflammatory cytokine IL-1b, 2, CINC-2 and CNTF in RV. - MGCD0103 ↓ RV stroke work and has No effect on RV hemodynamics (e.g. RVEDP, PRSW, and CO). - MS-275 ↓ RVH	22282194/[115]
**HDAC inhibitor Vorinostat**	N/A	- Calves, Male, Holstein dairy, 7 D (45–50 kg) - Hypobaric hypoxia (15,000 feet simulated elevation, PB = 430 mmHg, 14 days	N/A	suberanilohydroxamic acid, 5 mg/kg/day for 5 days	- HDAC1 increased in HX RV. - Vorinostat ↑ rv fraction area changes, and ↓ aSMA	34552503/[116]
**HDAC class 1** **HDAC class 2**	N/A	- Rats, Male, Sprague-Dawley, 250–280 g - PAB - MCT (60 mg/kg, s.c.) - SU5416 (20 mg/kg) + Hypoxia (10% O_2_, 3 W)	N/A	N/A	- MCT vs. ctrl RV=> ↑ HDAC Class I & II activity - SuHX vs. ctrl RV => ↑ HDAC Class I & II activity - PAB vs. ctrl RV => ↑ HDAC Class I & II activity	25006442/[114]
**Pressure** **overload**						
**HDAC inhibitors (TSA)**	N/A	- Rats, Male, Sprague-Dawley, 250–280 g - PAB	- HCMVEC - HCF	- In vivo TSA (IP, 450 μg/kg, 5/W, initiated 4 W post-surgery) - in vitro TSA (0.01–10 μM/24 H)	- TSA Does Not Prevent RVH - TSA Associated with RV Dysfunction, ↑ RV Fibrosis ↓ Capillary density - TSA => reactivation of fetal genes and ↑cardiomyocyte apoptosis	21297075/[118]
**HDAC inhibitor sodium valproate**	N/A	- Rats, Male, Sprague-Dawley, 3 W - PAB - MCT	N/A	- Sodium valproate (drinking water, 0.71% weight/vol, 3 weeks)	-Valproate ↓ RVH in both PAB and MCT rats. - Valproate ↓ fibrosis	20208383/[117]
**Systemic ventricle**						
**HDAC Class I, IIa, and IIb**	RV: - SV patients (*n* = 6, 17% female, 0.15 +/− 0.05 y), - non-failing control (*n* = 6, 17% female, 9.2 +/− 4.7 y)	- Rats, Sprague-Dawley - Neonatal hypoxia (10% O_2_, D1 for 1 W)	N/A	N/A	- ↑ HDAC Class I, IIa, and IIb catalytic activity and protein expression in human SV RV and HX neonatal rats - ↑ RVH in HX neonatal rats	28549058/[120]
**Tetralogy of Fallot**						
**HIRA**	RVOT: - 39 TOF patients - 4 noncardiac death children. Blood samples: - 100 TOF patients - and 200 healthy children	N/A	N/A	N/A	- ↓ HIRA mRNA and protein expressions In TOF Myocardium. - 5 SNP in *HIRA* promoter (g.4111A>G (rs1128399), g.4265C>A (rs4585115), g.4369T>G (rs2277837), g.4371C>A (rs148516780), and g.4543T>C (rs111802956))	27748330/[121]
**Arrhythmogenic cardio-myopathy**						
**H3K4me3**	N/A	N/A	AMRVM AMLVM	Genetic engenering NOTCH gain of function	- Notch activation => ↑H3K4me3 at the *Hrt2* promoter in the LV - Notch activation => ↑H3K4me3 at the *Hrt2* promoter in the LV	27697822/[111]
**GCN5**	RV - 16 ACM (41 y +/− 11.9) - 7 ctrl	N/A	CStCs	- GCN5 shRNA - GCN5 inhibitors MB-3	Human: - ACM vs. ctrl RV => ↑ *GCN5* In vitro - ACM vs. ctrl CStCs => ↑ *GCN5* - ↓ GCN5 => ↓ lipid accumulation, ↓ ROS in ACM CStCs	35712781/[119]

ACM: Arrhythmogenic Cardio-Myopathy; AMLVM: adult mouse left ventricular myocyte; AMRVM: adult mouse right ventricular myocyte; ANP: atrial natriuretic peptide, APJ: apelin receptor; ARVF: adult rat ventricular fibroblast; ARVM: adult rat ventricular myocyte; BNP: brain natriuretic peptide; CM: cardiomyocytes; CINC-2: cytokine-induced neutrophil chemoattractant-2; CNTF: Ciliary neurotrophic factor, and related; CO: cardiac output; CStCS: Cardiac stromal cells; ET1: endothelin-1; FBS: foetal bovine serum; H3/4: histone 3/4; HAND2: Heart- and neural crest derivatives-expressed protein 2; HAT: histone acetyl transferase; HCF: Human cardiac fibroblasts; HCMCEV: Human cardiac microvascular endothelial cells; HDAC: histone deacetylase; HMT: histone methyl transferase; HX: hypoxia; IL: interleukin; LV: left ventricle; MCT: monocrotaline; MEF2: myocyte enhancer factor 2; MHC: myosin heavy chain; NE: norepinephrine; NRVM: neonatal rat ventricular myocyte; OFT: outflow tract; PAB: pulmonary artery banding; PE: phenylephrine; PGF2A: platelet growth factor 2A; PRSW: Preload Recruitable Stroke Work; RV: right ventricle; RVEDP: right ventricular end diastolic pressure; RVH: right ventricular hypertrophy; RVOT: right ventricular outflow tract; SHD: second heart field; Smyd1: SET And MYND Domain Containing 1; SNP: single nucleotide polymorphism; SU5416: sugen; Su-Hx: sugen + hypoxia; SV: single ventricle; TSA: trichostatin A; TOF: tetralogy of Fallot.

### 3.11. DNA Methylation

DNA methylation is a biological process by which methyl groups are added to cytosine or adenine nucleotides. It is a dynamic process tightly regulated by the expression/activity of DNA methylation “writers”, which add methyl groups to the DNA (e.g., DNA methyl transferases 1, (DNMT1), DNMT3a, and DNMT3b), and the DNA methylation erasers, which remove methyl groups (e.g., the ten-eleven translocation (TET1, TET2, TET3)). Functionally, DNA methylation regulates gene expression by recruiting proteins involved in gene repression or by inhibiting the binding of transcription factor(s) to DNA. Although an impaired DNA methylation landscape has been associated with (left) heart failure, the contribution of DNA methylation to RV dysfunction remains poorly investigated [122] (Table 5).

Interestingly, the LV and the RV exhibit distinct DNA methylation profiles. RVs from human patients with DCM exhibited 1828 differentially methylated loci (compared to the LV), likely contributing to the regulation of cardiac development genes [123]. This observation suggests a tissue specificity in DNA methylation patterns. Consistently, blood leucocytes and right atrial tissues displayed distinct DNA methylation profiles in patients that underwent coronary artery bypass surgery [124]. In TOF RVOT or RV myocardium, an impaired DNA methylation landscape affected ETS1, RXR, and SP1 binding sites and contributed to decreased *DLK1*, *NOTCH4*, and *TBX20* gene expression [125,126,127,128,129,130]. Notably, hypomethylating drugs influenced the interaction between RXR, ETS1, and their respective binding sites in vitro [126,130]. Additionally, the impaired DNA methylation of *NKX2-5*, *EGFR*, *EVC2*, *TBX5*, *CFC1B*, *RXRa*, *VANGL2*, *TBX20*, *NR2F2*, *NOTCH4*, and *DLK1* correlated or was associated with the impaired expression of their respective transcripts [126,127,128,129,130,131,132,133,134]. Interestingly, the DNA methylation of TBX20 (affected in TOF patients) was unaffected in the RVs and blood of DCM patients, suggesting a potential specificity of DNA methylation changes for the pathologic condition affecting the RV [135,136]. The DNA methylation of *LINE-1* was decreased in RV samples from male TOF patients (compared to male controls) but not in females [125]. This observation supports the potential effect of sex on the DNA methylation landscape in TOF.

In PAH, DNA methylation contributes to pathological processes involved in RV dysfunction. Indeed, decompensated RV from PAH patients exhibited an increased expression of DNMT3a and DNMT3b, associated with increased DNA methylation of the angiomiR-126 gene, likely contributing to its downregulation and the angiogenic defect observed in patients’ RVs [56]. In vitro, the hypomethylating drug named hydralazine increased miR-126 and the angiogenic potential of endothelial cells isolated from PAH decompensated RVs. Moreover, RV fibroblasts harvested from a preclinical model of PH with RV failure exhibited an increased expression of DNMT1, compared to fibroblasts isolated from control (non-PH) animals [137]. In vitro, the hypomethylating drug decitabine decreased *DNMT1* expression and global DNA methylation, resulting in an improved fibroblast mitochondrial metabolism and a potentially decreased PH RV fibroblast pro-proliferative and pro-fibrotic phenotype. This observation echoes another study, which demonstrated that increased RV fibrosis was associated with *DNMT1* and *DNMT3a* overexpression in hypoxic mice [138]. In vitro, hypoxia increased DNA methylation, fibroblast proliferation, and fibrosis.

**Table 5 cells-12-02693-t005:** DNA methylation and right ventricle.

Epigenetic Modification	Human	In Vivo	In Vitro	Therapeutic Intervention	Outcome	PMID/Ref
**Congenital heart disease**						
**DNAm**	Blood - monozygotic twin with and without DORV (2 y, 100% female) - 15 DORV (60% male, 1–3.5 y) - 5 ctrl (60% male, 0.8–3.8 y)	N/A	N/A	N/A	- DORV vs. healthy twin => 1566 DMR, ↓ DNAm *ZIC3* and NR2F2 promoters - DORV vs. ctrl => ↓ *ZIC3* and *NR2F2* DNAm - *ZIC3* and *NR2F2* methylation negatively correlate with gene expression.	29866040/[136]
**Dilated cardiomyopathy**						
**- 450K Human-Methylation Array**	- 9 DCM RV free wall (57 +/− 6 yo) - 18 DCM LV apex (54 +/− 4 y)	N/A	N/A	N/A	- RV vs. LV => 544 hypermethylated, 1284 hypomethylated DMP. - Impaired DNAm enriched in the cis-regulatory regions of cardiac development genes	27417303/[123]
**DNAm (TBX20)**	RV (intraventricular septum) - 30 DCM (63% male) - 14 ctrl (64% male) PBMC - 150 DCM (41.9 y +/− 14.4) - 114 ctrl (42.9 y +/− 14.2)	N/A	N/A	N/A	- RV DCM vs. ctrl => no changes in *TBX20* promoter methylation - PBMC DCM vs. ctrl => no changes in *TBX20* promoter methylation	26895318/[135]
**Fibrosis**						
** - DNMT1** **- DNMT3a** **- DNAm**	N/A	- Mice, C57BL/6, male with extra copy for human EC-SOD - Hx (10% O_2_, 21 d)	- MCF	N/A	In Vivo: - Hx => ↑ RV DNMT1, DNMT3a, and fibrosis. - EC-SOD mice => ↓ DNMT1, DNMT3a, and RV fibrosis. In vitro: - Hx => ↑ DNAm in MCF - EC-SOD MCF => ↓ Hx induced DNAm - ↓ *RASSFA1* DNAm => ↓ MCF proliferation	34136546/[138]
**Pulmonary hypertension**						
**- DNMT1** **- DNAm** **- miR-148**	N/A	N/A	- RRVF - RLVF	5-Aza (10 μM, 48 h)	- MCT RRVF vs. Ctrl RRVF => ↑ DNMT1. - MCT RLVF vs. Ctrl RLVF => no difference in DNAm - 5-Aza => ↓ HIF-1a, ↑ mitochondrial metabolism in MCT-RRVF. - miR-148 predict to target DNMT1. - MCT RRVF => ↑ mir-148.	32216531/[137]
**-DNMT3a** **-DNMT3b** **- DNAm**	RV free wall tissue - 17 ctrl (62 +/− 4 years, 53% female) - 8 CRV (40 +/− 7 years, 50% female); - 14 PAH DRV (53 +/− 4 years 79% female)	N/A	- Human EC from ctrl RV, CRV and DRV	In vitro -Hydralazine	Human - ↑ DNMT3a and DNMT3B in DRV - ↑ DNAm of miR126 promoter in DRV In vitro - hydralazine => ↑ miR-126 expression => ↑ angiogenesis	26162916/[56]
**Surgery**						
**DNAm (b-MHC)**	Right atrial - 3 coronary artery bypass surgery Blood leukocytes - 3 coronary artery bypass surgery	N/A	N/A	N/A	- *b-MHC* promoter methylation negatively correlates with *b-MHC* mRNA expression	9747442/[124]
**Tetralogy of Fallot**						
**- DNAm (NKX2-5, GATA4, HAND1)**	RV myocardium: - 30 TOF (67% male, 1.13 +/− 0.85 y), - 6 ctrl (67% Male, 1.73 +/− 1.44 y)	N/A	N/A	N/A	- TOF vs. ctrl => ↑ *NKX2-5, HAND1* DNAm. - *NKX2-5* DNAm negatively correlates with mRNA levels.	24182332/[134]
**- Methyl Array**	RV myocardium - 41 TOF (63% male, 11 (7–24 m), - 6 ctrl (67% male, 15 (10.5–30 m)	N/A	N/A	N/A	- TOF vs. ctrl => 26 differentially methylated genes. - *EGFR, EVC2, TBX5, CFC1B* DNAm correlates with their respective mRNA	24479926/[133]
**- DNAm (*RXRa* promoter)**	RVOT: 6 TOF (69% male, 15.8 +/− 13 m), 6 ctrl (67% male, 21 +/− 17.3 m)	N/A	N/A	N/A	-TOF vs. ctrl => ↑ *RXRa* DNAm -TOF vs. ctrl => ↓ *RXRa* mRNA	24513686/[132]
**- DNAm (*VANGL2* promoter)**	RVOT: - 36 TOF (64% male, 11.5 (7.0–22.8 m) - 5 ctrl (60% male, 12 (2.0–42.0 m).	N/A	N/A	N/A	-TOF vs. ctrl => ↑ *VANGL2* DNAm -TOF vs. ctrl => ↓ *VANGL2* mRNA et protein	25200836/[131]
**- DNAm (TBX20)**	RVOT: - 23 TOF (48% male, 18 (11–35) m). - 5 ctrl (40% male, 19 (2.5–31 m).	N/A	N/A	N/A	- TOF vs. ctrl => ↓ *TBX20* DNAm - TOF vs. ctrl => ↑ *TBX20* mRNA - *TBX20* DNAm correlates with *TBX20* mRNA	30084275/[128]
**- DNAm (TBX20)**	RVOT: - 42 TOF (62% male, 1.28 +/− 1.05 y); - 6 ctrl (68% male, 1.73 +/− 1.44 y)	N/A	N/A	N/A	- TOF vs. ctrl => ↓ *TBX20* DNAm - DNAm affects TFBS for SP1. - SP1 might mediated increased *TBX20* expression.	31138201/[127]
**- DNAm (NR2F2)**	RVOT: - 25 TOF (60% male, 7.0 m (4.27–13)), - 5 ctrl (100% male, 2 m (0.14–9.5))	N/A	- HL-1 - HEK293T	- 5-Aza (5 µM, 48 h)	Human: TOF vs. Ctrl => ↓ *NR2F2* DNAm. *NR2F2* DNAm correlates with gene expression. In vitro: - DNAm affects RXRa binding site. - 5-Aza treatment influences RXRα affinity to its binding sites.	32819587/[126]
**- DNAm (NOTCH4)**	RVOT: - 24 TOF (58.3% male, 2.54 y +/− 0.86) - 5 ctrl (100% male, 0.35 y +/− 0.19)	N/A	N/A	N/A	- TOF vs. ctrl => ↑ *NOTCH4* DNAm - TOF vs. ctrl => ↓ *NOTCH4* mRNA - *NOTCH4* DNAm correlates *NOTCH4* expression - DNAm affect ETS1 binding sites	33000281/[129]
**- DNAm (DLK1)**	RVOT: - 25 TOF (60% male, 0.59 y (0.37–0.95)) - 5 ctrl (100% male, 0.17 y (0.01–0.79))	N/A	- HEK293T	- Decitabine	Humans: - TOF vs. ctrl => ↓ *DLK1* DNAm TOF - - TOF vs. ctrl => ↓ *DLK1* mRNA - NOTCH4 DNAm correlates *NOTCH4* expression - DNAm affect ETS1 binding sites - ETS1 inhibits *DLK1* expression. In Vitro: - Decitabine increased ETS1 binding.	35059744/[130]
**DNAm (LINE-1, NKX2-5, HAND1 and TBX 20)**	RV - 19 TOF (69% male, 19.8 m +/− 13.9) - 15 ctrl (66.7% male 13.4 y +/− 11.0 )	N/A	N/A	N/A	- TOF vs. ctrl => ↓ *LINE-1* DNAm, ↓ *TBX20* DNAm, ↑NKX2-5 and HAND1 - TOF male vs. ctrl male => => ↓ *LINE-1* DNAm (not in female) - *LINE-1* DNAm predict TOF.	22672592/[125]

5-Aza: 5-Azacytidine; CHD: Congenital heart diseases; CRV: compensated RV; Ctrl: control; DCM: dilated cardiomyopathy; DMP: differentially methylated probers; DNAm: DNA methylation; DNMT: DNA methyl transferase; DORV: double outlet RV; DRV: decompensated RV; HX: hypoxia; LV: left ventricle; MCF: mouse cardiac fibroblast; PH: pulmonary hypertension; PAH: pulmonary arterial hypertension; PBMC: peripheral blood monolayer cells; RLVF: rat LV fibroblast; RRVF: rat RV fibroblast; RV: right ventricle; RVOT: right ventricle outflow tract; SOD: superoxide dismutase; TOF: tetralogy of Fallot.

## 4. Conclusions

In comparison to other cardiovascular conditions, such as left heart failure and atherosclerosis, research on the role of epigenetic modifications in RV development, dysfunction, and failure is relatively sparse. To address this gap, we performed a thorough systematic review of the literature, focusing on studies that investigated the relationship between various epigenetic mechanisms (including DNA methylation, histone modification, and non-coding RNA) and RV pathophysiology. Our search on PubMed, up to 1 January 2023, yielded 109 original peer-reviewed articles.

Our review revealed that the bulk of the literature is descriptive in nature. A scant number of studies ventured into proposing epigenetic-based therapeutic strategies or biomarker development. This scarcity of interventional research could reflect the current limitations in pharmaceutical interventions that target epigenetic modifications, particularly those affecting DNA methylation and non-coding RNAs. The primary agents available, pan DNMT or HDAC inhibitors, induce widespread and non-selective effects, which are less than ideal.

However, the advent of sophisticated biomolecular techniques, such as the second generation of CRISPR/Cas9 systems, now permits precise DNA methylation editing at the single-nucleotide resolution [139] and targeted histone acetylation modifications at the single-gene level [140,141]. Despite their potential, these technologies are predominantly confined to preclinical studies. Additionally, devising RV-specific therapies that do not impact other organs (e.g., LV, lungs, and kidneys) poses a significant technical hurdle, thus impeding the progression to clinical trials for RV dysfunctions.

A further complication in the field is the inconsistency of study results, particularly concerning miRNA expression patterns in RV dysfunction. Conflicting data on whether RV dysfunction correlates with an upregulation or downregulation of specific miRNAs, such as miR-21, miR-223, or miR-146b, underscores the dynamic and intricate nature of epigenetic regulation throughout disease progression. It also highlights the necessity for standardized protocols in terms of control groups, epigenetic assessment technologies, and sample processing.

Moreover, the epigenetic underpinnings of various physiological processes affecting the RV, notably during the aging process, are poorly understood, and, consequently, the literature is very limited. For instance, similarly to the LV, aging markedly impairs RV function—this is manifested through reduced systolic and diastolic functions, increased fibrosis, cardiomyocyte death, and stiffness [142,143,144]—yet the molecular drivers of these age-related changes in the RV remain unclear, with no dedicated studies investigating their epigenetic foundations.

Interestingly, the RV exhibits a greater susceptibility to damage from intense physical exercise than the LV. Highly trained athletes, for example, display well-documented morphological and functional changes in the RV, such as decreased ejection fraction and increased wall stress, particularly after prolonged endurance activities [145,146,147,148]. These changes have been correlated with an elevated risk of atrial fibrillation. The causal factors behind RV dysfunction post-exercise are thought to be related to the ventricle’s unique physiological characteristics. Nonetheless, the precise molecular and potential epigenetic contributions to these changes are yet to be fully delineated.

In summary, studies have only just scratched the surface of the role of epigenetics in RV development and dysfunction, and despite promising results, additional investigations are needed to pinpoint the contribution of epigenetics in physiological and pathological RV development and the development of epigenetic-based therapy.

## Data Availability

Not applicable.

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
