# Peer review of "Right Ventricle and Epigenetics: A Systematic Review"

_cells, 2023, doi:10.3390/cells12232693_

Round 1
Reviewer 1 Report
Comments and Suggestions for Authors
Well worded abstract.
Great idea for this systemic review. The discussion can be improved with more robust literature search on the topic. Overall this article is great systemic review, great flow, well maintained structures and in my opinion acceptable to publish.
No further comments.
Author Response
Well worded abstract.
Great idea for this systemic review. The discussion can be improved with more robust literature search on the topic. Overall this article is great systemic review, great flow, well maintained structures and in my opinion acceptable to publish.
Thank you for your feedback. We have revised the discussion section, making an effort to retain its substance while ensuring brevity of the manuscript.

Reviewer 2 Report
Comments and Suggestions for Authors
In the review article ‘Right ventricle and epigenetics: A systematic review.’, the authors summarize the epigenetic knowledge about the right ventricle. The review article is interesting to many clinical and basic scientist. However, some parts must be improved before this review article should be accepted for publication:
1.) All human gene names (not protein names!) should be written in Italics!
2.) Please add relevant OMIM identifiers e.g. for ACM.
3.) Please explain why you have excluded review articles in your literature analysis? I can not understand this specific point, since many review articles contain and summarize important points. I would include some specific review articles if they fit into your story!
4.) In the introduction you should explain typical histological features of ACM and explain, why it was described in the older literature as ARVC. Recently, the group of Eva van Rooij investigate the fibro-fatty replacement in the myocardium using spatial transcriptomics (PMID 35576477). Since this landmark paper defines this fibro-fatty replacement at an impressive molecular detailed level, I would highly recommended to reference this paper in this context.
5.) In my view, you should also discuss the major ACM associated genes. I understand that your article is about the epigenetic changes and not directly about the genetic causes of ACM. However, I think that the readers need a little bit more information, which are the most relevant ACM associated genes. Especially, because you explain the regulation of desmocollin-2 by a specific miRNA. Please see and discuss the following genetic articles to ACM:
· PKP2 (encoding plakophilin-2) à Gerull B, et al. 2004 Mutations in the desmosomal protein plakophilin-2 are common in arrhythmogenic right ventricular cardiomyopathy.
· DSC2 (encoding desmocollin-2)à Brodehl A, et al. J Mol Cell Cardiol 2020 Apr:141:17-29. doi: 10.1016/j.yjmcc.2020.03.006. A homozygous DSC2 deletion associated with arrhythmogenic cardiomyopathy is caused by uniparental isodisomy.
· DSG2 (encoding desmoglein-2) à Pilichou K, et al. Circulation. 2006 Mar 7;113(9):1171-9. Mutations in desmoglein-2 gene are associated with arrhythmogenic right ventricular cardiomyopathy
· DSP (encoding desmoplakin) à Bauce B. et al. Eur Heart J 2005 Aug;26(16):1666-75. Clinical profile of four families with arrhythmogenic right ventricular cardiomyopathy caused by dominant desmoplakin mutations.
· DES (encoding desmin) à Bermúdez-Jiménez FJ et al. Circulation 2018 Apr 10;137(15):1595-1610. Novel Desmin Mutation p.Glu401Asp Impairs Filament Formation, Disrupts Cell Membrane Integrity, and Causes Severe Arrhythmogenic Left Ventricular Cardiomyopathy/Dysplasia.
6.) The abbreviation ACM was introduced two times. Please change this.
7.) I would generate schematic overview figures about the different epigenetic regulations in the different cardiac diseases of the right ventricle. Maybe Biorender could be a helpful software for this.
8.) I would introduce the Task Force Criteria for ACM diagnosis.
In summary, I suggest a major revision and advice the authors to give a little bit more details about ACM. Instead, it would be hard to be thrown into the epigenetic changes without explaining shortly the genetic basis of this disease. Good luck with the revision!
Comments on the Quality of English LanguageAn native speaking Editor should double check the English quality of this manuscript.
Author Response
Reviewer 2
In the review article ‘Right ventricle and epigenetics: A systematic review.’, the authors summarize the epigenetic knowledge about the right ventricle. The review article is interesting to many clinical and basic scientist. However, some parts must be improved before this review article should be accepted for publication:
Thank you for your input. Attached, please find a comprehensive response addressing each of your comments in detail.
- All human gene names (not protein names!) should be written in Italics!
In the revised manuscript and tables, gene names have been formatted in italics.
- Please add relevant OMIM identifiers e.g. for ACM.
We have included the relevant OMIM identifiers for specific genetic conditions where applicable. For instance, we referenced OMIM #610472 for desmocollin 2 in the context of Arrhythmogenic Cardiomyopathy (ACM).
- Please explain why you have excluded review articles in your literature analysis? I can not understand this specific point, since many review articles contain and summarize important points. I would include some specific review articles if they fit into your story!
Systematic reviews generally focus on primary research to ensure the novelty and originality of the analysis. In line with this standard, our review methodically selects original research articles and consciously omits review articles to preclude redundancy. Given that our systematic review already encompasses an extensive selection of 109 studies, we have a solid foundation for our decision to refine our inclusion criteria strictly to original, empirical research studies.
- In the introduction you should explain typical histological features of ACM and explain, why it was described in the older literature as ARVC. Recently, the group of Eva van Rooij investigate the fibro-fatty replacement in the myocardium using spatial transcriptomics (PMID 35576477). Since this landmark paper defines this fibro-fatty replacement at an impressive molecular detailed level, I would highly recommended to reference this paper in this context.
We appreciate your insightful comment. To address the concerns raised regarding ARVC, we have included the following statement in the section on ACM
“Arrhythmogenic cardiomyopathy (ACM) is a genetic disease characterized by a progressive fibro-fatty substitution of the ventricular myocardium, particularly pronounced in the RV, associated with life-threatening ventricular arrhythmias and, ultimately heart failure15. Traditionally, ACM was known as arrhythmogenic right ventricular cardiomyopathy (ARVC), stemming from the initial belief that the condition was confined exclusively to the RV chamber of the heart.”
While we acknowledge the reviewer's interest in the study by Eva van Rooij and colleagues, and recognize the impressive methodology it employs, the inclusion of this particular work does not align with the specific aims of our systematic review. Our focus is meticulously centered on cataloging epigenetic modifications associated with both the physiological and pathological development of the right ventricle, as well as disease progression. To maintain clarity and relevance within the scope of our review, which already encompasses a comprehensive analysis detailed in an 8000-word manuscript and includes 21 pages of tables, we have decided not to discuss this paper in depth. However, in deference to the reviewer's recommendation, we have cited the paper within our review for readers seeking additional insights
- In my view, you should also discuss the major ACM associated genes. I understand that your article is about the epigenetic changes and not directly about the genetic causes of ACM. However, I think that the readers need a little bit more information, which are the most relevant ACM associated genes. Especially, because you explain the regulation of desmocollin-2 by a specific miRNA. Please see and discuss the following genetic articles to ACM:
- PKP2 (encoding plakophilin-2) à Gerull B, et al. 2004 Mutations in the desmosomal protein plakophilin-2 are common in arrhythmogenic right ventricular cardiomyopathy.
- DSC2 (encoding desmocollin-2)à Brodehl A, et al. J Mol Cell Cardiol 2020 Apr:141:17-29. doi: 10.1016/j.yjmcc.2020.03.006. A homozygous DSC2 deletion associated with arrhythmogenic cardiomyopathy is caused by uniparental isodisomy.
- DSG2 (encoding desmoglein-2) à Pilichou K, et al. Circulation. 2006 Mar 7;113(9):1171-9. Mutations in desmoglein-2 gene are associated with arrhythmogenic right ventricular cardiomyopathy
- DSP (encoding desmoplakin) à Bauce B. et al. Eur Heart J 2005 Aug;26(16):1666-75. Clinical profile of four families with arrhythmogenic right ventricular cardiomyopathy caused by dominant desmoplakin mutations.
- DES (encoding desmin) à Bermúdez-Jiménez FJ et al. Circulation 2018 Apr 10;137(15):1595-1610. Novel Desmin Mutation p.Glu401Asp Impairs Filament Formation, Disrupts Cell Membrane Integrity, and Causes Severe Arrhythmogenic Left Ventricular Cardiomyopathy/Dysplasia.
Thank you for your valuable feedback. Consistent with the aims of this systematic review, which is to analyze epigenetic modifications linked to right ventricular (RV) development and dysfunction, we concentrate on the role of epigenetic regulation within the pathophysiology of various RV-related conditions, such as arrhythmogenic cardiomyopathy (ACM), coronary artery disease, congenital heart diseases, and pulmonary hypertension. While the genetic foundations of these conditions are indeed pertinent, our review maintains a focused scope, presenting a succinct overview of disease pathophysiology in conjunction with an analysis of epigenetic changes associated with RV dysfunction across these diseases. To accommodate your suggestion, we have enriched the introductory section on ACM with the recommended citations, thereby providing a pathway for readers to explore the genetic dimensions of ACM in depth.
- The abbreviation ACM was introduced two times. Please change this.
Done
- I would generate schematic overview figures about the different epigenetic regulations in the different cardiac diseases of the right ventricle. Maybe Biorender could be a helpful software for this.
We initially considered enriching our manuscript with figures to illustrate the epigenetic modifications. Nonetheless, given the extensive array of modifications identified (e.g. over 60 miRNAs implicated in RV dysfunction) the resulting figures became overly complex and difficult to interpret. Acknowledging the starkness of tables, we maintain that they offer the clearest, most comprehensive means of presenting the intricate landscape of epigenetic changes that play a role in RV dysfunction.
- I would introduce the Task Force Criteria for ACM diagnosis.
Acknowledging the broad focus of this systematic review beyond ACM alone, we have deliberately provided succinct disease descriptions to center attention on the primary subject: the spectrum of epigenetic modifications related to RV dysfunction across a variety of diseases. Nevertheless, in an effort to address the reviewer's suggestion and guide readers interested in a more detailed exploration, we have incorporated the suggested reference (PMID: 37844667) into the revised manuscript.

Reviewer 3 Report
Comments and Suggestions for Authors
This manuscript “Right ventricle and epigenetics: A systematic review” presents a review on the studies of epigenetic modifications associated with RV function/dysfunction.
The authors conducted this review based on numerous studies including that that used human samples. The outcome of this review is very important to understand the link between the epigenetic regulation of gene expression and RV development, RV physiological function, and RV pathological dysfunction.
However, it would be important if authors could include the available information for gene expression for idiopathic PH, heritable PH, PH that associated with various conditions including connective tissue diseases, HIV infection, portal hypertension, and congenital heart disease.
This manuscript may be considered for publishing in Cells with major changes.
Author Response
This manuscript “Right ventricle and epigenetics: A systematic review” presents a review on the studies of epigenetic modifications associated with RV function/dysfunction.
The authors conducted this review based on numerous studies including that that used human samples. The outcome of this review is very important to understand the link between the epigenetic regulation of gene expression and RV development, RV physiological function, and RV pathological dysfunction.
However, it would be important if authors could include the available information for gene expression for idiopathic PH, heritable PH, PH that associated with various conditions including connective tissue diseases, HIV infection, portal hypertension, and congenital heart disease.
Thank you for your insightful comment. The aim of our systematic review is to offer a detailed synthesis of the existing knowledge concerning epigenetic modifications and their influence on the physiological and pathological evolution of right ventricular (RV) function. In pursuit of this goal, we employed a rigorous systematic review methodology, underpinned by a selection of keywords that meticulously intersect the domains of the right ventricle and epigenetics. During our literature search, any articles that were not captured by our strategic approach or failed to meet our inclusion criteria were excluded.
Thus, regarding the gene expression patterns across different pulmonary hypertension (PH) groups, such articles were considered outside our review's eligibility criteria. While we acknowledge the potential interest and scientific merit in exploring the unique transcriptomes associated with each PH group, such an investigation would be expansive and merits a dedicated research article of its own. Moreover, as indicated earlier, our review does not solely concentrate on pulmonary hypertension but rather encompasses a broader spectrum of RV-related diseases. Consequently, delineating specific transcriptomes for each condition represented would present a considerable challenge and lies outside the confines of our current review's scope.

Reviewer 4 Report
Comments and Suggestions for Authors
Toro et al. presented an interesting review of accumulated evidence on the epigenetic regulation of gene expression in relation to different aspects of RV development, its function in physiological states but also in pathological conditions, both referring to congenital, inherited and acquired conditions.
Authors described very precisely methodology, and their strategy to pick up all relevant published studies from inception of PubMed. 109 studies were included in the review, and Authors report what is known on the role of microRNAs, long noncoding RNAs, histone modifications and DNA methylation across the most important disease states.
Authors conclude that the literature on the epigenetic modification in the RV development, dysfunction and failure is limited, and often there is the discrepancy between studies regarding epigenetic modifications, especially with regard to the role of miRNAs, in particular miR-21, miR223 or miR-146b. Consequently, there are no clinical trials assessing epigenetic-based therapy for RV dysfunction.
A major setback of this review is a lack of recognition that the right ventricle in contradistinction to the left ventricle is more prone to damage during training. There is no address to the issue of enhanced right ventricle remodeling in response to endurance training in marathon runners or elite athletes.
Another important setback is a lack of data on the RV changes during ageing.
This data should be complemented in the review, or at least the info included and conclusions should be modified accordingly.
Minor remarks
The definiton of dilated cardiomyopathy (American or European) includes hypokinesis of the LV or both ventricles but rarely showing cutoff criteria in terms of LVEF. It is commonly shown in original research papers just to show who was included in the study either patients with LVEF<50% ( with milder LV dysfunction also) or with more severe LV dysfunction (LVEF<40%). So, it should be changed in accordance to new European guidelines. PMID:37622657
Also with newer European guidelines on the management of cardiomyopathies the notion of arrhythmogenic cardiomyopathy disappeared, and ARVC relates as previously to arrhythrogenic right ventricular cardiomyopathy.
In the description of Pulmonary hypertension Authors twice recognize group 3 PH both due to left heart disease and due to CTPEH.
The notion sugen should be explained.
Results related to instead of no, there are few other spelling mistakes.
Comments on the Quality of English Language
Very few spelling mistakes, the text is diffcult but understadable.
Author Response
Toro et al. presented an interesting review of accumulated evidence on the epigenetic regulation of gene expression in relation to different aspects of RV development, its function in physiological states but also in pathological conditions, both referring to congenital, inherited and acquired conditions.
Authors described very precisely methodology, and their strategy to pick up all relevant published studies from inception of PubMed. 109 studies were included in the review, and Authors report what is known on the role of microRNAs, long noncoding RNAs, histone modifications and DNA methylation across the most important disease states.
Authors conclude that the literature on the epigenetic modification in the RV development, dysfunction and failure is limited, and often there is the discrepancy between studies regarding epigenetic modifications, especially with regard to the role of miRNAs, in particular miR-21, miR223 or miR-146b. Consequently, there are no clinical trials assessing epigenetic-based therapy for RV dysfunction.
A major setback of this review is a lack of recognition that the right ventricle in contradistinction to the left ventricle is more prone to damage during training. There is no address to the issue of enhanced right ventricle remodeling in response to endurance training in marathon runners or elite athletes.
Another important setback is a lack of data on the RV changes during ageing.
This data should be complemented in the review, or at least the info included and conclusions should be modified accordingly.
Thank you for your astute observation. The central theme of our systematic review is to elucidate the role of epigenetic modifications in the regulation of both physiological and pathological development and function of the RV. The changes in the RV associated with aging and physical exertion were not addressed in our initial manuscript draft, primarily due to the lack of studies that have investigated these aspects within the scope of epigenetic.
We concur with the reviewer that the intersection of epigenetic modifications with RV changes due to aging and exertion represents a compelling topic for further exploration. Accordingly, we have included an additional paragraph in the discussion section to acknowledge this intriguing area of research. This inclusion aims to highlight the significance of these factors, to underscore the current gaps in our understanding, and to suggest directions for future research that might illuminate the epigenetic landscape influencing RV adaptation in these contexts.
“Moreover, the epigenetic underpinnings of various physiological processes affecting the RV, notably during the aging process, are poorly understood and consequently the literature very limited. For instance, similar to the LV, aging markedly impairs RV function—manifested by reduced systolic and diastolic functions, increased fibrosis, cardiomyocyte death, and stiffness138-140—yet the molecular drivers of these age-related changes in the RV remain unclear, with no dedicated studies investigating their epigenetic foundations.
Interestingly, the RV exhibits a greater susceptibility to damage from intense physical exercise than the LV. Highly trained athletes, for example, display well-documented morphological and functional changes in the RV, such as decreased ejection fraction and increased wall stress, particularly after prolonged endurance activities141-144. These changes have been correlated with an elevated risk of atrial fibrillation. The causal factors behind RV dysfunction post-exercise are thought to be related to the ventricle's unique physiological characteristics. Nonetheless, the precise molecular and potential epigenetic contributions to these changes are yet to be fully delineated."
Minor remarks
The definiton of dilated cardiomyopathy (American or European) includes hypokinesis of the LV or both ventricles but rarely showing cutoff criteria in terms of LVEF. It is commonly shown in original research papers just to show who was included in the study either patients with LVEF<50% ( with milder LV dysfunction also) or with more severe LV dysfunction (LVEF<40%). So, it should be changed in accordance to new European guidelines. PMID:37622657
It has been changed and we included the suggested reference.
Also with newer European guidelines on the management of cardiomyopathies the notion of arrhythmogenic cardiomyopathy disappeared, and ARVC relates as previously to arrhythrogenic right ventricular cardiomyopathy.
It has been clarified according to the comment from reviewer 2 point 4.
In the description of Pulmonary hypertension Authors twice recognize group 3 PH both due to left heart disease and due to CTPEH.
corrected
The notion sugen should be explained.
We clarified that Sugen is a VEGFR blocker
Results related to instead of no, there are few other spelling mistakes.
Manuscript has been reviewed for typos.

Round 2
Reviewer 2 Report
Comments and Suggestions for Authors
Congratulations! The authors have significantly improved their manuscript. I suggest to accept this manuscript for publication in Cells!
Reviewer 3 Report
Comments and Suggestions for Authors
The authors stated that the aim of their systematic review is detailed synthesis of the existing knowledge concerning epigenetic modifications and their influence on the physiological and pathological evolution of right ventricular (RV) function. This manuscript may be considered for publishing in the Cells.